# NICEdrug.ch, a workflow for rational drug design and systems-level analysis of drug metabolism

Homa MohammadiPeyhani, Anush Chiappino-Pepe[†‡], Kiandokht Haddadi[†§], Jasmin Hafner[#], Noushin Hadadi[¶], Vassily Hatzimanikatis*

Laboratory of Computational Systems Biotechnology, École Polytechnique Fédérale de Lausanne, EPFL, Lausanne, Switzerland

*For correspondence:
vassily.hatzimanikatis@epfl.ch

[†]These authors contributed equally to this work

Present address: [‡]Department of Genetics, Harvard Medical School, Boston, United States; [§]Department of Chemical Engineering and Applied Chemistry, University of Toronto, Toronto, Canada; [#]Department of Environmental Chemistry, Eawag, Switzerland; [¶]Department of Cell Physiology and Metabolism, Université de Genève, Geneva, Switzerland

Competing interests: The authors declare that no competing interests exist.

## Abstract

The discovery of a drug requires over a decade of intensive research and financial investments – and still has a high risk of failure. To reduce this burden, we developed the NICEdrug.ch resource, which incorporates 250,000 bioactive molecules, and studied their enzymatic metabolic targets, fate, and toxicity. NICEdrug.ch includes a unique fingerprint that identifies reactive similarities between drug–drug and drug–metabolite pairs. We validated the application, scope, and performance of NICEdrug.ch over similar methods in the field on golden standard datasets describing drugs and metabolites sharing reactivity, drug toxicities, and drug targets. We use NICEdrug.ch to evaluate inhibition and toxicity by the anticancer drug 5-fluorouracil, and suggest avenues to alleviate its side effects. We propose shikimate 3-phosphate for targeting liver-stage malaria with minimal impact on the human host cell. Finally, NICEdrug.ch suggests over 1300 candidate drugs and food molecules to target COVID-19 and explains their inhibitory mechanism for further experimental screening. The NICEdrug.ch database is accessible online to systematically identify the reactivity of small molecules and druggable enzymes with practical applications in lead discovery and drug repurposing.

## Introduction

To assure effective therapies for previously untreated illness, emerging diseases, and personalized medicine, new small molecules are needed. However, the process to develop new drugs is complex, costly, and time consuming. This is especially problematic considering that about 90% of drug candidates in clinical trials are discarded due to unexpected toxicity or other secondary effects. This inefficiency threatens our health care system and economy (*Wong et al., 2019*). Improving how we discover and design new drugs could reduce the time and costs involved in the developmental pipeline and hence is of primary importance to define efficient medical therapies.

Current drug discovery techniques often involve high-throughput screens with candidates and a set of target enzymes presumably involved in a disease, which leads to the selection for those candidates with the preferred activity. However, the biochemical space of small molecules and possible targets in the cell is huge, which limits the possible experimental testing. Computational methods for drug pre-screening and discovery are therefore promising. In silico, one can systematically search the maximum biochemical space for targets and molecules with desired structures and functions to narrow down the molecules to test experimentally.

There are two main in silico strategies for drug discovery: a data-driven approach based on machine learning or a mechanistic approach based on the available biochemical knowledge. Machine learning (ML) has been successfully used in all stages of drug discovery, from the prediction of targets to the discovery of drug candidates, as shown in some recent studies (*Reker et al., 2020*; *Shilo et al., 2020*; *Stokes et al., 2020*; *Vamathevan et al., 2019*). However, ML approaches require

big, high-quality data sets of drug activity and associated physiology (*Vamathevan et al., 2019*), which might be challenging to obtain when studying drug action mechanisms and side effects in humans. ML also uses trained neural networks, which can lack interpretability and repeatability. This can make it difficult to explain why the neural networks has chosen a specific result, why it unexpectedly failed for an unseen dataset, and the final results may vary (*Vamathevan et al., 2019*).

Mechanistic-based approaches can also rationally identify small molecules in a desired system and do not require such large amounts of data. Such methods commonly screen based on structural similarity to a native enzyme substrate (anti-metabolite) or to a known drug (for drug repurposing), considering the complete structure of a molecule to extract information about protein–ligand fitness (*Jarvis and Ouvry, 2019*; *Verlinde and Hol, 1994*). However, respecting enzymatic catalysis, the reactive sites, and neighboring atoms play a more important role than the rest of the molecule when assessing molecular reactivity (*Hadadi et al., 2019*). Indeed, reactive-site-centric information might allow to identify (1) the metabolic fate and neighbors of a small molecule (*Javdan et al., 2020*), including metabolic precursors or prodrugs and products of metabolic degradation, (2) small molecules sharing reactivity (*Lim et al., 2010*), and (3) competitively inhibited enzymes (*Ghattas et al., 2016*). Furthermore, neither ML nor mechanistic-based approaches consider the metabolism of the patient, even though the metabolic fate of the drug and the existence of additional targets in the cell might give rise to toxicity. To our knowledge, no available method accounts for human biochemistry when refining the search for drugs.

In this study, we present the development of the NICEdrug.ch database using a more holistic and updated approach to a traditional mechanistic-based screen by (1) adding a more detailed analysis of drug molecular structures and enzymatic targets based on structural aspects of enzymatic catalysis and (2) accounting for drug metabolism in the context of human biochemistry. NICEdrug.ch assesses the similarity of the reactivity between a drug candidate and a native substrate of an enzyme based on their common reactive sites and neighboring atoms (i.e., the NICEdrug score) in an analogous fashion as the computational tool BridgIT (*Hadadi et al., 2019*). It also identifies all biochemical transformations in the cellular metabolism that can modify and degrade a drug candidate using a previously developed reaction-prediction tool, termed Biochemical Network Integrated Computational Explorer (BNICE.ch) (*Hatzimanikatis et al., 2005*; *Soh and Hatzimanikatis, 2010*) and the ATLAS of Biochemistry (*Hadadi et al., 2016*; *Hafner et al., 2020*). With NICEdrug.ch, we automatically analyzed the functional, reactive, and physicochemical properties of around 250,000 small molecules to suggest the action mechanism, metabolic fate, toxicity, and possibility of drug repurposing for each compound.

To prove the predictive power of NICEdrug.ch in large-scale analysis, we collected and tested over 70,000 drug–enzyme pair inhibition data from available bioassays and high-throughput screening studies. Our comparison of predicted and experimentally tested drug–enzyme pairs shows that NICEdrug.ch predictive accuracy is over 70%. Remarkably, half of the drugs in this comparison show 100% accuracy. We have listed five potential sources of disagreement for the remaining half (accuracy of 65%) including drugs acting through non-competitive inhibition, which are out of the scope of NICEdrug.ch. Moreover, we have evaluated the accuracy of NICEdrug.ch predictions on drugs and metabolites that share reactivity, drug toxicity, and drug targets using golden standard datasets, i.e., a set of experimentally observed drug metabolites (*Flynn et al., 2020*; *Kirchmair et al., 2015*), a collection of cytotoxicity bioassay records from PubChem (*Svensson et al., 2017*; *Webel et al., 2020*; *Yin et al., 2019*), and a collection of drug–protein interactions reported in PubChem bioassays (*Kim et al., 2021*; *Wang et al., 2012*).

We apply NICEdrug.ch to study drug action mechanisms and identify drugs for repurposing related to four diseases: cancer, high cholesterol, malaria, and COVID-19. We also sought for molecules in food, as available in fooDB the largest database of food constituents (*Scalbert et al., 2011*), with putative anti SARS-CoV-2 activity. Finally, we provide NICEdrug.ch as an online resource (https://lcsb-databases.epfl.ch/pathways/Nicedrug/). Overall, NICEdrug.ch combines knowledge of molecular structures, enzymatic reaction mechanisms (as included in BNICE.ch; *Finley et al., 2009*; *Hadadi and Hatzimanikatis, 2015*; *Hatzimanikatis et al., 2005*; *Henry et al., 2010*; *Soh and Hatzimanikatis, 2010*; *Tokic et al., 2018*), and cellular biochemistry (currently human, *Plasmodium*, and *Escherichia coli* metabolism) to provide a promising and innovative resource to accelerate the discovery and design of novel drugs.

## Results

### NICEdrug.ch discovers 200,000 bioactive molecules one reaction away from known drugs in a human cell

To build the initial NICEdrug.ch database, we gathered over 70,000 existing small molecules presumed suitable for treating human diseases from three source databases: KEGG, ChEMBL, and DrugBank (*Figure 1—figure supplement 1*, method). We eliminated duplicate molecules, curated available information, computed thermodynamic properties, and applied the Lipinski rules (*Lipinski et al., 2001*) to keep only the molecules that have drug-like properties in NICEdrug.ch (*Figure 1*, 'Materials and methods'). NICEdrug.ch currently includes 48,544 unique small molecules from the source databases.

To evaluate the reactivity of the 48,544 drugs and drug candidates, we searched for all possible reactive sites on each molecule with BNICE.ch (*Hatzimanikatis et al., 2005*; *Figure 1*, 'Materials and methods'). All of the 48,544 molecules contain at least one reactive site and hence might be reactive in a cell. In total, we identified more than 5 million potential reactive sites (183 k unique) on the 48,544 molecules and matched them to a corresponding enzyme by assigning them to an Enzyme Commission (EC) number. All of these enzymes belong to the human metabolic network (*Supplementary file 1*, 'Materials and methods'). Interestingly, 10.4% of identified reactive sites correspond to the p450 class of enzymes, which are responsible for breaking down compounds in the human body by introducing reactive groups on those compounds, also known as phase I of drug metabolism (*Figure 1—figure supplement 2A*). The sites that were identified varied greatly from simple and small (i.e., comprising a minimum number of one atom) to more complex sites that covered a large part of the molecule. The biggest reactive site includes 30 atoms (*Figure 1—figure supplement 2B*).

Given the important role of metabolism in the biochemical transformations and toxicity of drugs (*Dumoulin et al., 2020*), we investigated the metabolism of the 48,544 input molecules in human cells. We predicted the hypothetical biochemical neighborhoods of all NICEdrug.ch small molecules in a human cell (i.e., reacting with known human metabolites and cofactors) using a retro-biosynthetic analysis with BNICE.ch (*Figure 1—figure supplement 1*, 'Materials and methods'). With this approach, we discovered 197,246 unique compounds connected to the input drugs and drug candidates via one step or reaction (products of the first generation), and the associated hypothetical biochemical neighborhood consists of 630,449 reactions (*Figure 1—figure supplement 2*). The 197,246 unique compounds are part of a new set of bioactive molecules in NICEdrug.ch that might act as drugs or prodrugs in a human cell. We stored the total number of 245,790 small molecules (including the curated set of 48,544 drugs and drug candidates and the new set of 197,246 bioactive compounds), their calculated properties, and biochemistry in our open-access database of drug metabolism, NICEdrug.ch.

To use NICEdrug.ch to identify drug-drug or drug–metabolite pairs that have shared reactivity and target enzymes, we developed a new metric called the *NICEdrug score* (*Figure 1—figure supplement 3*). The NICEdrug score uses information about the structure of the reactive site and its surroundings (as computed using the BridgIT methodology) and is stored in the form of a fingerprint ('Materials and methods'). The fingerprint of a molecule's reactive site and the neighborhood around this reactive site—termed the *reactive site-centric fingerprint*—serves to compare this site-specific similarity with other molecules. We recently showed that the reactive site-centric fingerprint of a reaction provides a better predictive measure of similar reactivity than the overall molecular structure, as the overall structure can be much larger than the reactive site and skew the results by indicating high similarities when the reactivity is actually quite different (*Hadadi et al., 2019*). Here, we generated reactive site-centric fingerprints for all 20 million reactive sites identified in the 48,544 drug–drug candidates and 197,246 one-step-away molecules included in NICEdrug.ch. The 20 million reactive site-centric fingerprints for the total 245,790 small molecules are available in NICEdrug.ch to be used in similarity comparisons and classifying molecules ('Materials and methods').

We propose the usage of NICEdrug.ch to generate reports that define the hypothetical reactivity of a molecule, the molecule's reactive sites as identified by target enzymes, and the NICEdrug score between drug–drug and drug–metabolite pairs. The NICEdrug.ch reports can be used for three main applications: (1) to identify the metabolism of small molecules; (2) to suggest drug repurposing; and (3) to evaluate the druggability of an enzyme in a desired cell or organism (*Figure 1*), as we

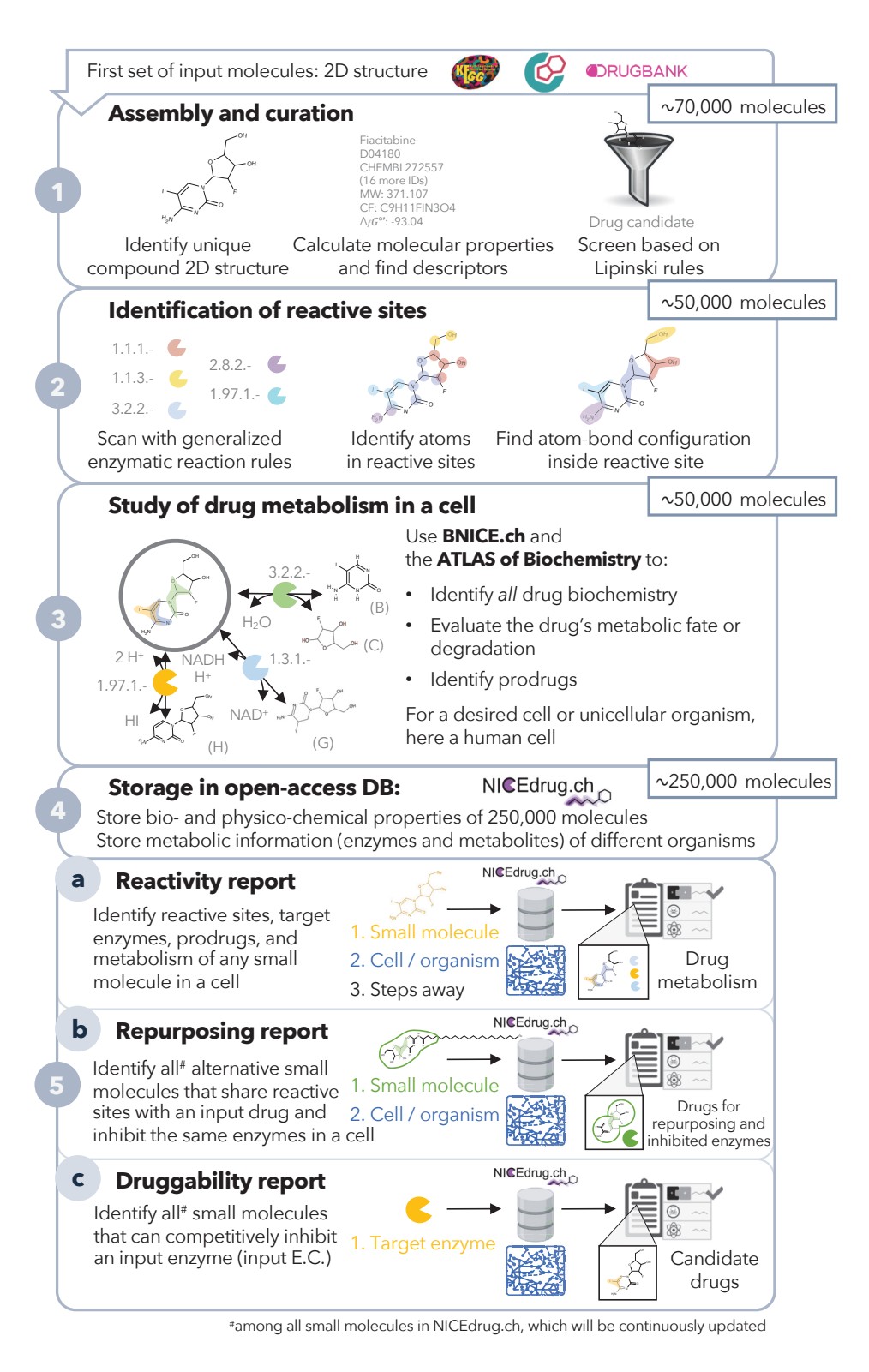

**Figure 1.** Pipeline to construct and use the NICEdrug.ch database. NICEdrug.ch (1) curates available information and calculates the properties of an input compound; (2) identifies the reactive sites of that compound; (3) explores the hypothetical metabolism of the compound in a cell; (4) stores all functional, reactive, bio-, and physico-chemical properties in open-source database; and (5) allows generation of reports to evaluate (5a) reactivity of a

*Figure 1 continued on next page*

*Figure 1 continued*

small molecule, (5b) drug repurposing, and (5c) druggability of an enzymatic target. See also  *Figure 1—figure supplement 1*; *Figure 1—figure supplement 2*; *Figure 1—figure supplement 3*,  and *Supplementary file 1* .

The online version of this article includes the following figure supplement(s) for figure 1:

**Figure supplement 1.** Overview of number of molecules in NICEdrug.ch and their structural curation.

**Figure supplement 2.** Distribution of reactive sites and metabolic reactions as of EC numbers linked to all molecules in NICEdrug.ch.

**Figure supplement 3.** Description of NICEdrug score.

show in the next sections. Currently, NICEdrug.ch includes metabolic information for human cells, a malaria parasite, and *Escherichia coli*, and it is easily extendible to other organisms in the future.

## Validation of NICEdrug.ch against biochemical assays

To prove the potential of NICEdrug.ch to predict the druggability (through competitive inhibition) of an enzyme by a small molecule, we compare a set of 70 k NICEdrug.ch drug–enzyme pair predictions with available biochemical assays and high-throughput compound screenings (*Supplementary file 2*, 'Materials and methods'). The set of 70 k drug–enzyme pairs involves all available active and inactive inhibition data for 2570 small molecules and 198 enzymes in the Pub-Chem Bioassays database (*Wang et al., 2012*). A comparison between the drugs' predicted and measured bioactivity against enzymes results in a predictive accuracy of NICEdrug.ch of 0.73. Interestingly, we identify two clusters of drugs: a set of 1269 small molecules for which the NICEdrug.ch predictions are 100% accurate and a set of 1301 drugs with 65% accuracy. We investigated the reasons for the mismatches and identify five explanations (*Supplementary file 2*, 'Materials and methods').

We have also compared the scope and application of NICEdrug.ch and other available computational drug discovery tools (*Supplementary file 2*), and we show how NICEdrug.ch outperforms the scope and predictive potential of all these tools (*Supplementary file 2*, 'Materials and methods').

We also quantitatively compared the accuracy of NICEdrug.ch predictions over similar methods in the field using golden standard datasets. These experimental datasets describe drugs and metabolites that share reactivity, drug toxicities, and drug targets. Hence, they serve to evaluate the NICEdrug.ch reactivity, druggability, and repurposing reports.

### Evaluation of NICEdrug.ch reactivity report

To evaluate the NICEdrug.ch reactivity report, we first used an experimental set including 29 small molecules and their 55 unique metabolic products (labeled in public databases) (*Flynn et al., 2020*). We compared the predictive accuracy of NICEdrug.ch with other tools predicting reactivity, i.e., XenoNet (*Flynn et al., 2020*), GLORY (*de Bruyn Kops et al., 2021*; *de Bruyn Kops et al., 2019*), SyGMa (*Ridder and Wagener, 2008*), and BioTransformer (*Djoumbou-Feunang et al., 2019*). NICE-drug.ch predicted 53 of the 55 metabolic products from the small molecule dataset, rendering a sensitivity score of 0.96. The two metabolites missing are venetoclax and SCHEMBL18637099, which are produced through at least one reaction with an unknown reaction mechanism and hence are out of the scope of NICEdrug.ch. The tools XenoNet, GLORY, SyGMa, and BioTransformer showed a sensitivity score of 0.89, 0.83, 0.74, and 0.72 on the same dataset. To this end, not only NICEdrug.ch outperforms previous tools, but it also provides information on the metabolic pathways and reaction mechanisms involved in the production of each metabolic product (see 'Materials and methods', *Supplementary file 2*). We next evaluated the NICEdrug.ch reactivity with a second dataset including 16 pairs of drugs and metabolite that share reactivity (*Kirchmair et al., 2015*) ('Materials and methods', *Supplementary file 2*). NICEdrug.ch correctly identified the pathways metabolizing 15 of the drugs and the associated metabolites sharing reactivity (*Supplementary file 2*).

### Evaluation of NICEdrug.ch toxicity report

As done before (*Svensson et al., 2017*; *Webel et al., 2020*; *Yin et al., 2019*), we used cytotoxicity bioassay records from PubChem (*Svensson et al., 2017*) involving 1777 drugs to evaluate the NICE-drug.ch toxicity report. Other available tools predict drug toxicity using machine learning. The accuracy of the machine-learning-based methods ranged from 0.67 to 0.78, as previously reported

(*Svensson et al., 2017*; *Yin et al., 2019*). For the same dataset, NICEdrug.ch shows an accuracy of 0.94, with a precision, recall, and F1 of 0.94, 0.92, and 0.96, respectively ('Materials and methods', *Supplementary file 2*).

## Evaluation of NICEdrug.ch druggability report

We compared the NICEdrug.ch druggability report with the widely used 'network-based inference (NBI)' tool for drug–target interaction (DTI) prediction. As a basis for this comparison, we used the high-quality drug–enzyme bioassay data from PubChem (*Kim et al., 2021*; *Wang et al., 2012*), which includes 651 records reporting the inhibition of 78 enzymes by 297 molecules. The area under the curve (AUC), a commonly used criterion for assessing computational target prediction methods (*Mayr et al., 2018*), quantified a remarkable improvement in the overall performance of NICEdrug. ch (0.85) over the NBI tool (0.61). Further analysis found that the optimal druggability scores is 0.46, with precision, recall, and F1 values of 0.88, 0.89, and 0.89, respectively ('Materials and methods', *Supplementary file 2*).

## NICEdrug.ch suggests inhibitory mechanisms of the anticancer drug 5-FU and avenues to alleviate its toxicity

As a case study, we used NICEdrug.ch to investigate the mode of action and metabolic fate of one of the most commonly used drugs to treat cancer, 5-fluorouracil (5-FU), by exploring its reactivity and the downstream products or intermediates that are formed during the cascade of biochemical transformations. 5-FU interferes with DNA synthesis as an anti-metabolite (*Longley et al., 2003*), meaning that its various intermediates like 5-fluorodeoxyuridine monophosphate (FdUMP) are similar enough to naturally occurring substrates and they can act as competitive inhibitors in the cell.

We therefore used NICEdrug.ch to study the intermediates of 5-FU that occurred between one to four reaction steps away from 5-FU (*Supplementary file 3*), which is a reasonable range to occur in the body after 5-FU treatment (*Testa, 2010*). This analysis identified 407 compounds (90 biochemical and 317 chemical molecules) that have the biochemical potential to inhibit certain enzymes. Because the NICEdrug score that analyses reactive site and neighborhood similarities can serve as a better predictor of metabolite similarity, we assessed the NICEdrug score of the intermediates compared to human metabolites. This resulted in a wide range of NICEdrug scores between the different 5-FU intermediates and human metabolites, ranging from no similarity at a NICEdrug score of 0 to the equivalent substructure on a compound at a NICEdrug score of 1. More importantly, some of the 407 metabolite inhibitors (as explained next) were known compounds that have been investigated for their effects on 5-FU toxicity, but most of these compounds were newly identified by NICEdrug.ch and could therefore serve as avenues for future research into alleviating the side effects of this drug.

We investigated these 407 compounds in more detail, looking first at the set of already validated metabolite inhibitors. 5-Fluorouridine (two steps away from 5-FU) and UDP-L-arabinofuranose (four steps away from 5-FU) are very similar to uridine, with NICEdrug scores of 0.95 and 1, respectively. Uridine is recognized as a substrate by two human enzymes: cytidine deaminase (EC: 3.5.4.5) and 5′-nucleotidase (EC: 3.1.3.5) (*Figure 2*). Therefore, NICEdrug.ch predictions show that the degradation metabolism of 5-FU generates downstream molecules similar to uridine, which likely leads to the inhibition of these two enzymes. This effect has already been investigated as a potential method for reducing the toxicity of 5-FU, wherein it was proposed that high concentrations of uridine could compete with the toxic 5-FU metabolites (*Ma et al., 2017*).

NICEdrug.ch also identified a few potential metabolites that have not been previously studied for their effects. These metabolites share a reactive site with native human metabolites and differ in the reactive site neighborhood, and we refer to them as *para-metabolites* (*Sartorelli and Johns, 2013*). 6-Methyl-2′-deoxyadenosine, purine-deoxyribonucleoside, and 2′-deoxyisoguanosine structurally resemble the reactive site neighborhood of deoxyadenosine, with respective NICEdrug scores of 1, 1, and 0.91. Similarly, 2-aminoadenosine, 2-chloroadenosine, and 2-methylaminoadenosine (four steps from 5-FU) have the same reactive site neighborhood as adenosine, with NICEdrug scores of 1, 1, and 0.96, respectively. Adenosine and deoxyadenosine are both native substrates of the adenosine kinase (EC: 2.7.1.20) and 5′-nucleotidase (EC: 3.1.3.5) (*Figure 2*). Therefore, we suggest that the 5-FU derivatives 2-aminoadenosine and 2-chloroadenosine are competitive inhibitors for the two

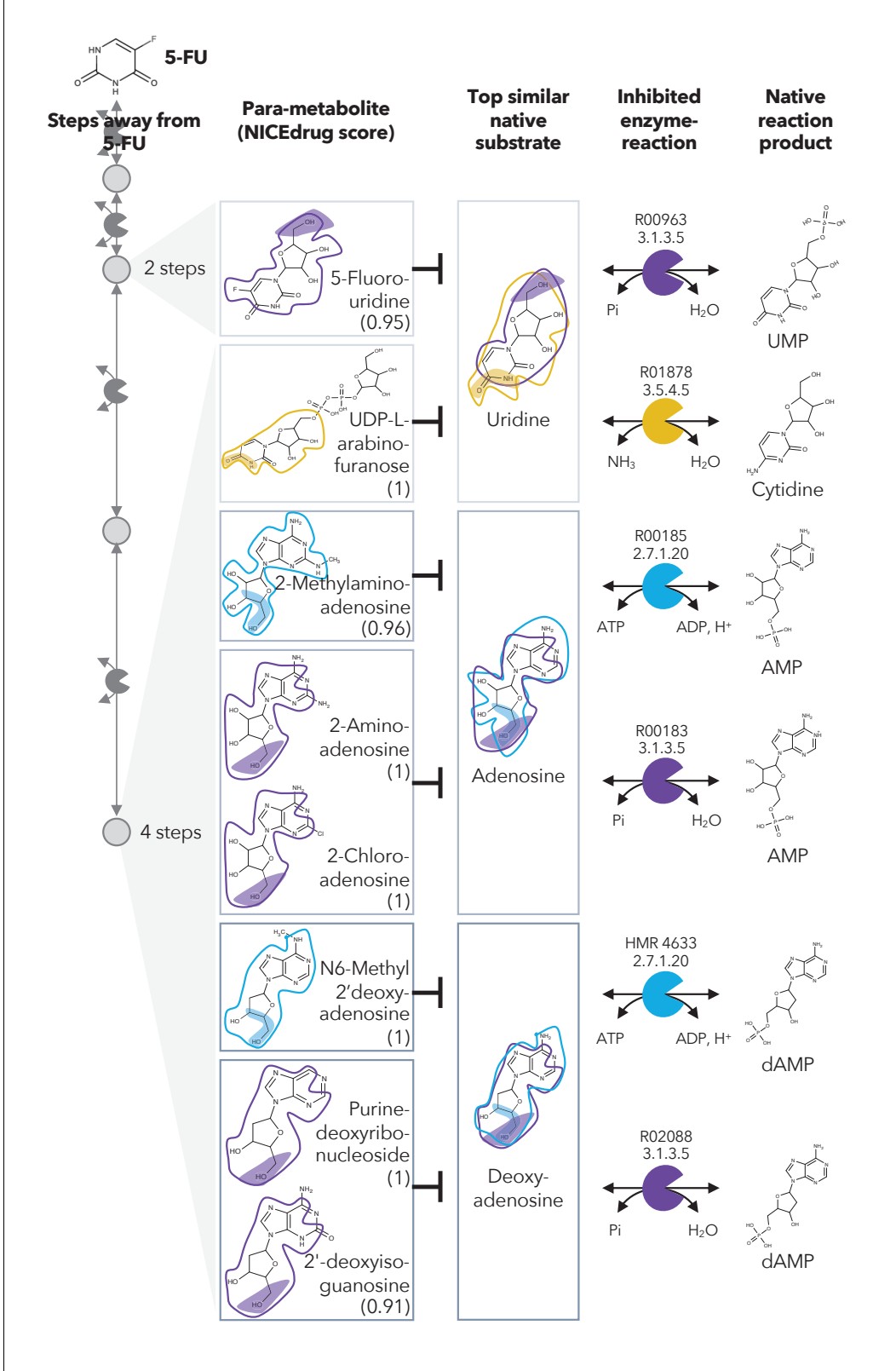

**Figure 2.** Similarity in reactive site and neighborhood defines para-metabolites in 5-FU metabolism and inhibited human metabolic enzymes. Eight para-metabolites in the 5-FU metabolic neighborhood (represented as defined in 'Materials and methods'). We show the most similar native human metabolites, inhibited enzymes, and native products of the reactions. See also *Supplementary file 3* .

enzymes adenosine kinase and 5′-nucleotidase. With these new insights from NICEdrug.ch, we hypothesize that co-administering adenosine or deoxyadenosine and uridine (*Figure 2*) with 5-FU might be required to reduce its toxic effects and hopefully alleviate the side effects of the 5-FU cancer treatment.

## Metabolic degradation of 5-FU leads to compounds with Fluor in their reactive site that are less reactive and more toxic than other intermediates

In the previous case study, we showed inhibitors that contain the identical active site to the native enzyme. However, a slightly different reactive site might still be able to bind to an enzyme and compete with a native substrate, also defined as *anti-metabolite* (*Matsuda et al., 2014*). We explored this scenario by defining relaxed constraints in two steps. We first identified all atoms around a reactive site to compare the binding characteristics between the native molecule and putative inhibitor. Next, we compared the reactive site of the native molecule and putative inhibitor and scored the latter based on similarity ('Materials and methods'). Following these two steps, we assessed the similarity between intermediates in the 5-FU metabolic neighborhood and human metabolites. Among all 407 compounds in the 5-FU metabolism (*Supplementary file 3*), we found eight that show a close similarity to human metabolites (NICEdrug score above 0.9, *Figure 3*) that might be competitive inhibitors or anti-metabolites. Inside the reactive site, the original hydrogen atom is bioisosterically replaced by fluorine. F–C bonds are extremely stable and therefore block the active site by forming

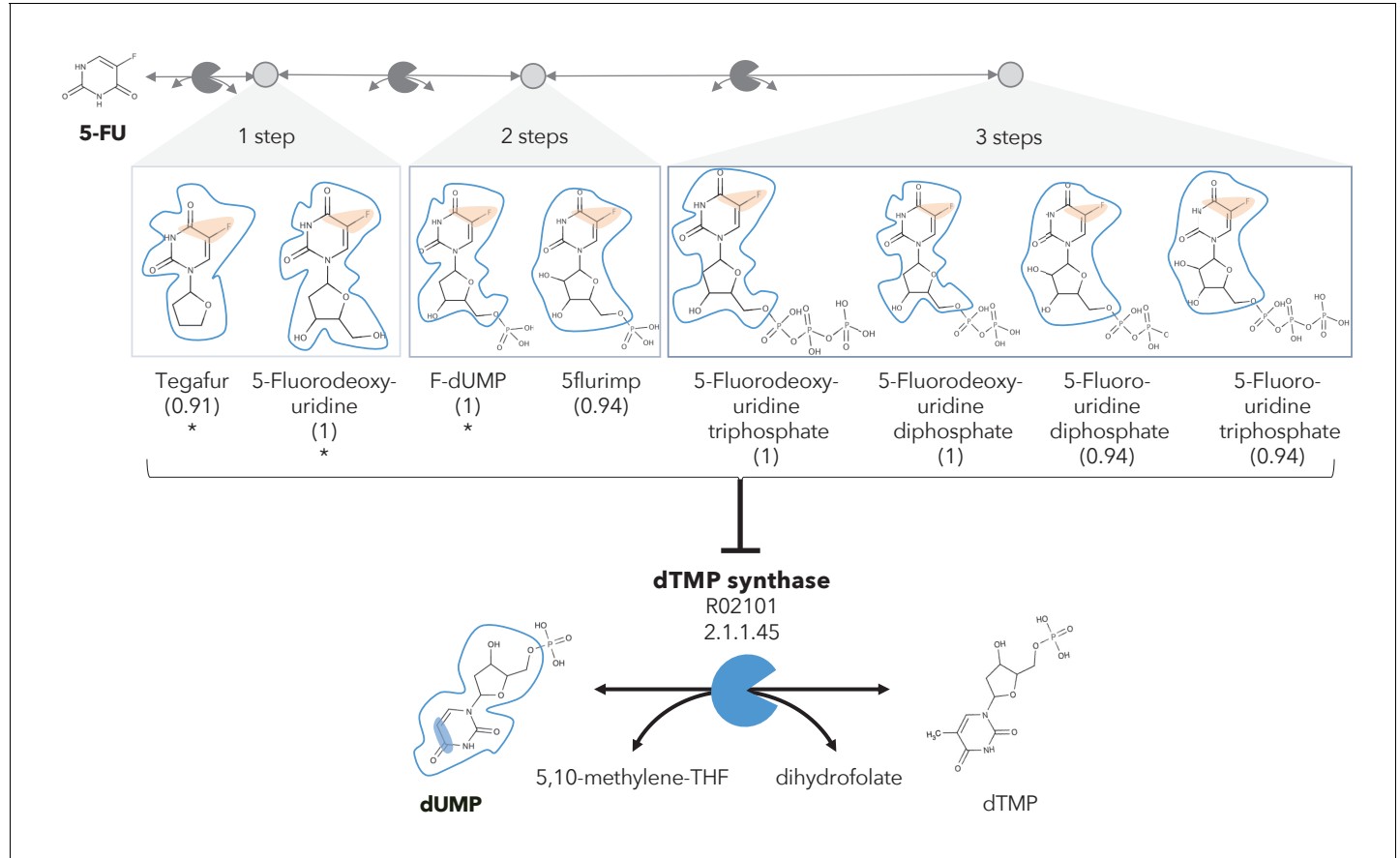

**Figure 3.** A different reactive site but similar neighborhood defines top anti-metabolites in 5-FU metabolism and inhibited human metabolic enzyme. Eight anti-metabolites of dUMP in the 5-FU metabolic neighborhood (represented as defined in 'Materials and methods'). Note that the reactive site of the anti-metabolites is different than the one of the native human metabolite, but the neighborhood is highly similar, which determines the high NICEdrug score (value in parenthesis). We show the inhibited human enzyme (dTMP synthase) and reaction, and its native product. See also *Supplementary file 3*

a stable complex with the enzyme. The inhibitory effect of the intermediates tegafur, 5-fluorodeoxyuridine, and FdUMP (one to two reaction steps away) has been confirmed in studies by *Kobayakawa and Kojima, 2011* and *Bielas et al., 2009*. In addition, NICEdrug.ch also predicts that 5flurim, 5-fluorodeoxyuridine triphosphate, 5-fluorodeoxyuridine triphosphate, 5-fluorouridine diphosphate, and 5-fluorouridine triphosphate, some of which occur further downstream in the 5-FU metabolism, also act as anti-metabolites (*Figure 3*). Based on the insights from NICEdrug.ch, we suggest the inhibitory and side effect of 5-FU treatment might be more complex than previously thought. 5-FU downstream products are structurally close to human metabolites and might form stable complexes with native enzymes. This knowledge could serve to further refine the pharmacokinetic and pharmacodynamic models of 5-FU and ultimately the dosage administered during treatment.

## NICEdrug.ch identifies toxic alerts in the anticancer drug 5-FU and its products from metabolic degradation

The concept of drug toxicity refers not to overdoses but instead to the toxic effects at medical doses (*Guengerich, 2011*), which often occur due to the degradation products generated through drug metabolism. Extensive efforts have been expended to identify toxic molecules or, more generally, to extract the substructures that are responsible for toxicity (called structural alerts). The Liver Toxicity Knowledge Base (LTKB) and the super toxic database include 1036 and about 60 k toxic molecules, respectively (*Schmidt et al., 2009*; *Thakkar et al., 2018*). ToxAlert provides around 1200 alerts related to different forms of toxicity (*Sushko et al., 2012*). However, the number of molecules that are analyzed and labeled as toxic in databases is disproportionally low compared to the space of compounds. Additionally, structural alerts are indicated for many compounds, and current alerts might identify redundant and over-specific substructures, which questions their reliability (*Yang et al., 2017*).

To quantify the toxicity of downstream products of drugs in NICEdrug.ch, we collected all of the molecules cataloged as toxic in the LTKB and super toxic databases (approved toxic molecules) along with their lethal dose ($LC_{50}$), as well as the existing structural alerts provided by ToxAlert. We measured the similarity of an input molecule with all approved toxic molecules using the reactive site-centric fingerprints implemented in BridgIT and the NICEdrug score ('Materials and methods'). Next, we scanned both the toxic reference molecule and the input molecule for structural hints of toxicity, referred to here as *NICEdrug toxic alerts*. We kept common NICEdrug toxic alerts between the reference, which is a confirmed toxic compound, and input molecule. With this procedure in place, NICEdrug.ch finds for each input molecule the most similar toxic molecules, along with their common toxic alerts, and serves to assess the toxicity of a new molecule based on the mapped toxic alerts. Additionally, the NICEdrug toxic alerts and toxicity level of drug intermediates can be traced with NICEdrug.ch through the whole degradation pathway to reveal the origin of the toxicity.

As an example, we herein tested the ability of NICEdrug.ch to identify the toxicity in 5-FU metabolism. First, we queried the toxicity profile of all intermediates in the 5-FU metabolic neighborhood, integrating both known and hypothetical human reactions ('Materials and methods'). In this analysis, we generated all compounds up to four steps away from 5-FU. Based on the toxicity report of each potential degradation product, we calculated a relative toxicity metric that adds the $LC_{50}$ value, NICEdrug score, and number of common NICEdrug toxic alerts with all approved toxic drugs ('Materials and methods'). We generated the metabolic neighborhood around 5-FU and labeled each compound with our toxicity metric (*Supplementary file 3*). Interestingly, we show that the top most toxic intermediates match the list of known three toxic intermediates in 5-FU metabolism (*Figure 4*; *Krauß and Bracher, 2018*). Based on the toxicity analysis in NICEdrug.ch for 5-FU, we hypothesize there are highly toxic products of 5-FU drug metabolism that had not been identified either experimentally or computationally and it might be necessary to experimentally evaluate their toxicity to recalibrate the dosage of 5-FU treatment.

## The nicedrug.ch reactive site-centric fingerprint accurately clusters statins of type I and II and guides drug repurposing

Because potential side effects of a drug are documented when the drug passes the approval process, repurposing approved drugs for other diseases can reduce the medical risks and development

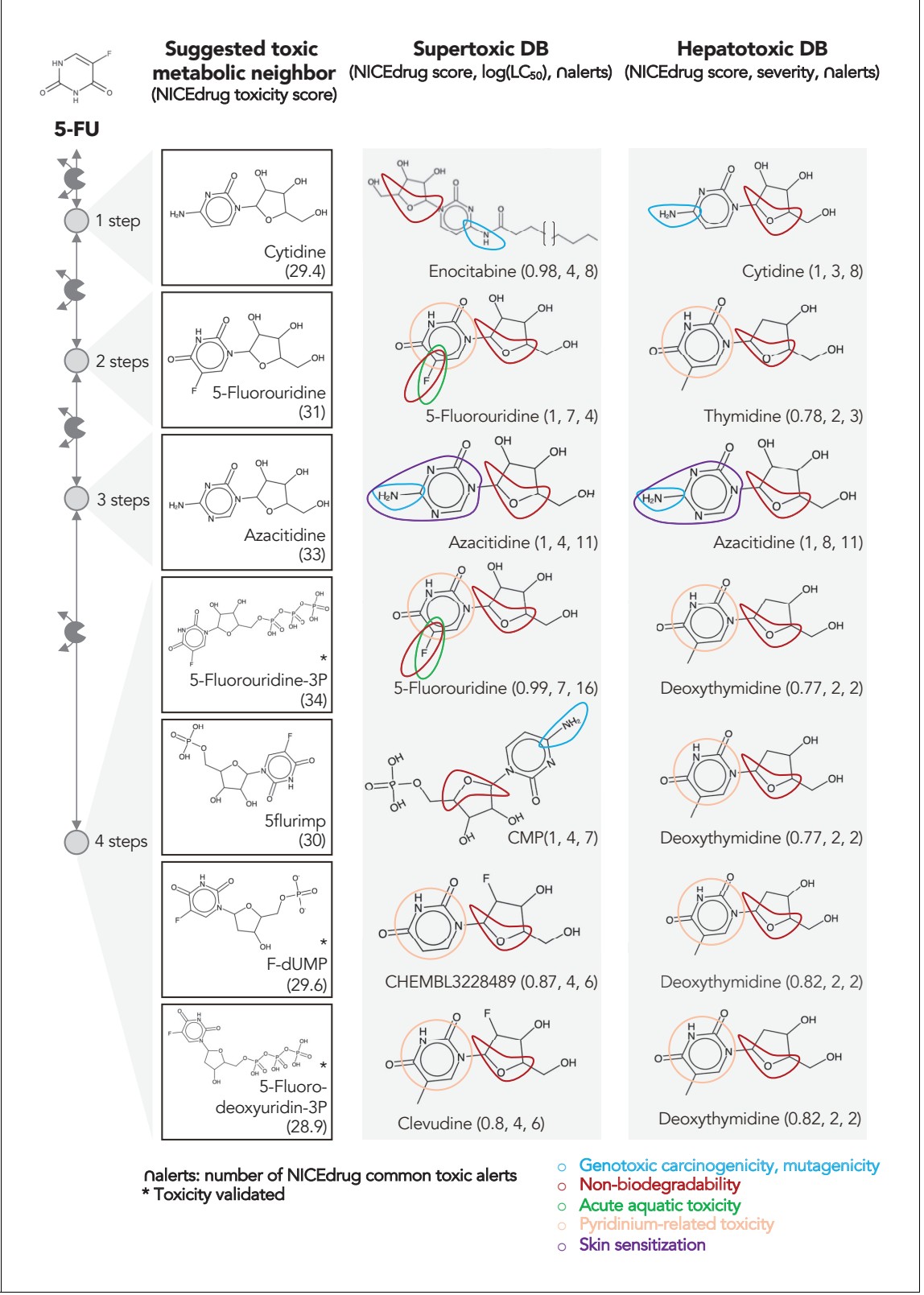

**Figure 4.** Comparing downstream products to known toxic molecules and analyzing their common structural toxic alerts explains metabolic toxicity of 5-FU. Example of six suggested toxic molecules in the 5-FU metabolic neighborhood (represented as defined in 'Materials and methods'). We show toxic compounds from the supertoxic and hepatotoxic databases that lead to the highest NICEdrug toxicity score (number under toxic intermediate

*Figure 4 continued on next page*

*Figure 4 continued*

name, 'Materials and methods'). We highlight functional groups linked to five NICEdrug toxic alerts (legend bottom right). See also *Supplementary file 3*.

expenses (*Himmelstein et al., 2017*). For instance, the antitussive noscapine has been repurposed to treat some types of cancers (*Mahmoudian and Rahimi-Moghaddam, 2009*; *Rajesh, 2011*). Because NICEdrug.ch can search for functional (i.e., reactivity), structural (i.e., size), and physicochemical (i.e., solubility) similarities between molecules while accounting for human biochemistry, we wanted to determine whether NICEdrug.ch could therefore suggest drug repurposing strategies.

As a case study, we investigated the possibility of drug repurposing to replace statins, which are a class of drugs often prescribed to lower blood cholesterol levels and to treat cardiovascular disease. Indeed, data from the National Health and Nutrition Examination Survey indicate that nearly half of adults 75 years and older in the United States use prescription cholesterol-lowering statins (*Bibbins-Domingo et al., 2016*). Since some patients do not tolerate these drugs and many still do not reach a safe blood cholesterol level (*Kong et al., 2004*), there is a need for alternatives. Being competitive inhibitors of the cholesterol biosynthesis enzyme 3-hydroxy-3-methyl-glutaryl-coenzyme A reductase (HMG-CoA reductase) (*Jiang et al., 2018*; *Mulhaupt et al., 2003*), all statins share the same reactive site. BNICE.ch labeled this reactive site, in a linear or circular form, as corresponding to an EC number of 4.2.1.- (*Istvan and Deisenhofer, 2001*). NICEdrug.ch includes 254 molecules with the same reactive site that are recognized by enzymes of EC class 4.2.1.-, ten of which are known statins. We used the NICEdrug score to cluster the 254 molecules into different classes (*Supplementary file 4*, *Figure 5*). Two of the classes correspond to all currently known statins, which are classified based on their activity into types 1 and 2, wherein statins of type two are less active and their reactive site is more stable compared to type 1. This property is well distinguished in the clustering based on the NICEdrug score (*Figure 5A*).

In addition to properly classifying the 10 known statins (*Figure 5B,C*, molecules non-marked), we identified seven other NICEdrug.ch molecules that clustered tightly with these statins (*Figure 5B,C*, molecules marked with \*). These new molecules share the same reactive site and physicochemical properties, and they have the highest similarity with known statins in atoms neighboring the reactive site. In a previous study by *Endo and Hasumi, 1993*, these seven NICEdrug.ch molecules were introduced as Mevastatin analogues for inhibiting cholesterol biosynthesis. Therefore, they were already suggested as possible candidates for treating high blood cholesterol and could be a good option for repurposing. Furthermore, we found eight known drugs not from the statin family among the 254 scanned molecules (*Supplementary file 4*). One of them, acetyl-L-carnitine (*Figure 5C*, molecule marked with \*\*), is mainly used for treating neuropathic pain (*Li et al., 2015*), though *Tanaka et al., 2004* have already confirmed that it also has a cholesterol-reducing effect.

Overall, NICEdrug.ch was able to characterize all known enzymatic reactions that metabolize statins, including proposed alternatives and new hypothetical reactions that could be involved in their metabolism within human cells (*Figure 5A*, *Figure 5—figure supplement 1*). The identification of seven drugs that clustered around the statins and were already designed as alternatives to statins confirms the power of NICEdrug.ch and the NICEdrug score to search large databases for similar compounds in structure and function. Furthermore, the discovery of the eight compounds unrelated to known statins offer multiple candidate drugs for repurposing along with a map of their metabolized intermediates for the treatment of high cholesterol, though further preclinical experiments would be required to verify their clinical benefits.

## NICEdrug.ch suggests over 500 drugs and drug candidates to target liver-stage malaria and simultaneously minimize side effects in human cells, with shikimate 3-phosphate as a top candidate

Efficiently targeting malaria remains a global health challenge. Malaria parasites (*Plasmodium*) are developing resistance to all known drugs, and antimalarials cause many side effects (*World Health Organization, 2018*). We applied NICEdrug.ch to identify drug candidates that target liver-stage developing malaria parasites and lessen or avoid side effects in human cells.

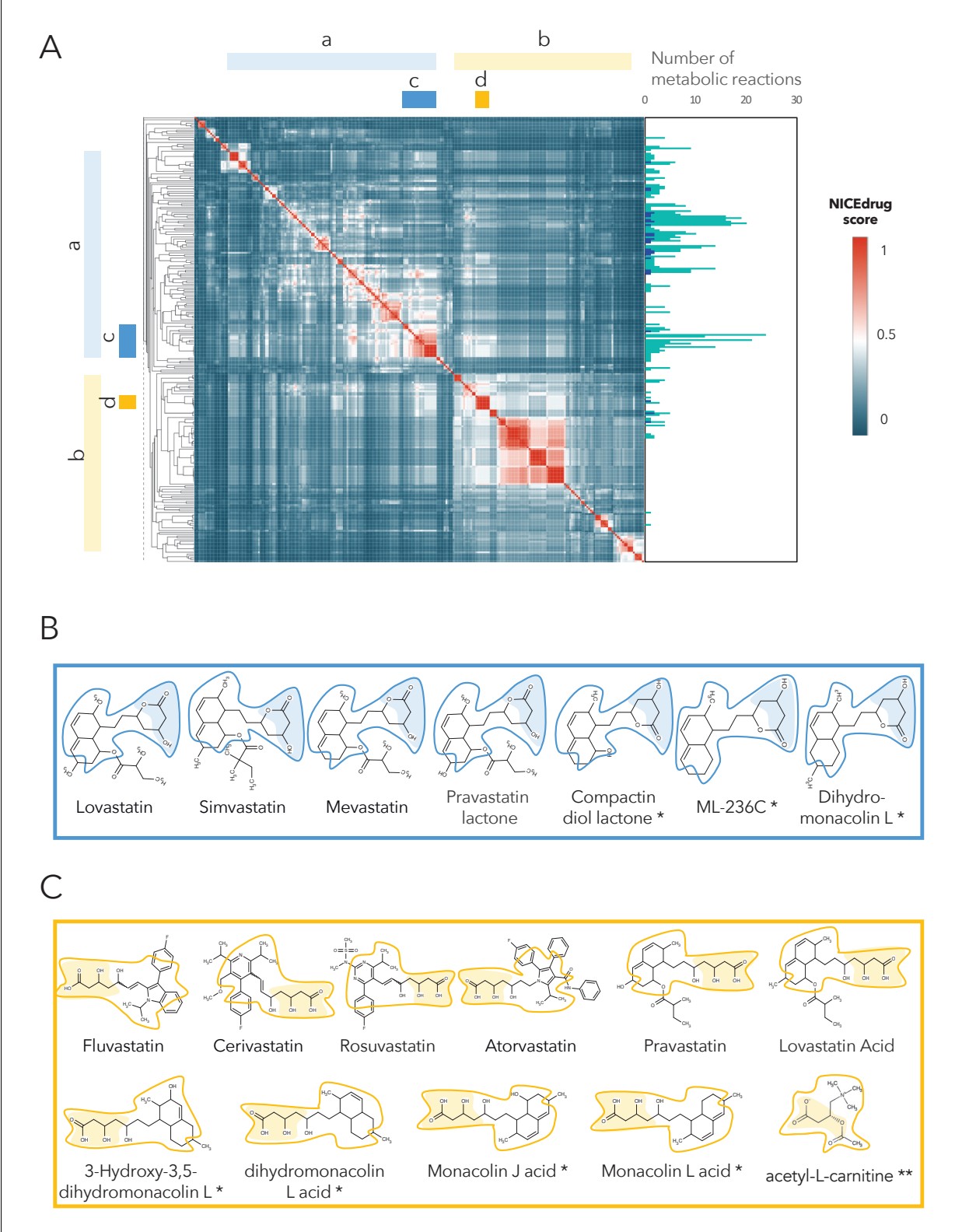

**Figure 5.** Clustering of molecules with statin reactive sites based on NICEdrug score suggests drugs for repurposing. (**A**) Pairwise NICEdrug score between all molecules with statin reactive sites (heat map) and number of metabolic reactions in which they participate (right). We highlight clusters of statins of type 1 (cluster a) and type 2 (cluster b), and clusters of most similar molecules to type one statins (cluster c) and type two statins (cluster d). Within the metabolic reactions, we indicate the total number of reactions (dark color) and the number of reactions that involve the statin reactive site

*Figure 5 continued on next page*

(light color). (B) Examples of statins and Mevastatin analogues of type one from cluster c (blue) and of type two from cluster d (gold). We left the known statins unmarked, which are appropriately clustered together based on the NICEdrug score, and we mark with * new molecules that cluster with statins and that NICEdrug.ch suggests could be repurposed to act as statins. Reactive sites in type one statins and type two statins are colored in blue and orange, respectively. The reactive site neighborhood as considered in the NICEdrug score is also marked. See also; *Figure 5—figure supplement 1* , and *Supplementary file 4*.

The online version of this article includes the following figure supplement(s) for figure 5:

**Figure supplement 1.** Clustering based on NICEdrug score, molecular weight, and reactivity of statin like molecules.

We previously reported 178 essential genes and enzymes for liver-stage development in the malaria parasite *Plasmodium berghei* (*Stanway et al., 2019*; *Supplementary file 5*, 'Materials and methods'). Of 178 essential *Plasmodium* enzymes, 32 enzymes are not essential in human cells (*Wang et al., 2015*; *Supplementary file 5*, 'Materials and methods'). We extracted all molecules catalyzed by these 32 enzymes uniquely essential in *Plasmodium*, which resulted in 68 metabolites and 157 unique metabolite–enzyme pairs (*Supplementary file 5*, 'Materials and methods'). We used NICEdrug.ch to examine the druggability of the 32 essential *Plasmodium* enzymes with the curated 48,544 drugs and drug candidates (*Figure 1*) and the possibility of repurposing them to target malaria.

We considered as candidates for targeting liver-stage malaria as the drugs or their metabolic neighbors that show a good NICEdrug score (NICEdrug score above 0.5) with any of the 157 *Plasmodium* metabolite–enzyme pairs. We identified 516 such drug candidates, targeting 16 essential *Plasmodium* enzymes (*Supplementary file 6*, 'Materials and methods'). Furthermore, 1164 other drugs appear in the metabolic neighborhood of the 516 identified drugs (between one and three reaction steps away). Interestingly, of the 516 identified drug candidates, digoxigenin, estradiol-17beta, and estriol have been previously validated as antimalarials (*Antonova-Koch et al., 2018*), and NICEdrug.ch suggests their antimalarial activity relies on the competitive inhibition of the KRC enzyme (*Figure 6*). This enzyme is part of both the steroid metabolism and the fatty acid elongation metabolism, which we recently showed is essential for *Plasmodium* liver-stage development (*Stanway et al., 2019*). Among the 516 NICEdrug.ch antimalarial candidates, there are also 89 molecules present in the metabolic neighborhood of antimalarial drugs approved by *Antonova-Koch et al., 2018*, which suggests these antimalarials might be prodrugs (*Supplementary file 6*).

Being an intracellular parasite, antimalarial treatments should be efficient at targeting *Plasmodium* as well as assure the integrity of the host cell (*Figure 6A*). To tackle this challenge, we identified 1497 metabolites participating in metabolic reactions catalyzed with essential human enzymes (*Supplementary file 5*, 'Materials and methods') and excluded the antimalarial drug candidates that shared reactive site-centric similarity with the extracted human metabolite set (to satisfy NICEdrug score below 0.5). Of all 516 drug candidates that might target liver-stage *Plasmodium*, a reduced set of 64 molecules minimize the inhibition of essential human enzymes (*Supplementary file 6*, 'Materials and methods') and are hence optimal antimalarial candidates.

Among our set of 64 optimal antimalarial candidates, a set of 14 drugs targeting the *Plasmodium* shikimate metabolism, whose function is essential for liver-stage malaria development (*Stanway et al., 2019*), arose as the top candidate because of its complete absence in human cells. The set of drug candidates targeting shikimate metabolism include 40 prodrugs (between one and three reaction steps away) that have been shown to have antimalarial activity (*Antonova-Koch et al., 2018*; *Supplementary file 6*). NICEdrug.ch identified molecules among the prodrugs with a high number of toxic alerts, like nitrofen. It also identified four molecules with scaffolds similar (two or three steps away) to the 1-(4-chlorobenzoyl)pyrazolidin-3-one of shikimate and derivatives. This result suggests that downstream compounds of the 40 prodrugs might target the *Plasmodium* shikimate pathway, but also might cause side effects in humans (*Supplementary file 6*).

To this end, NICEdrug.ch identified shikimate 3-phosphate as a top candidate antimalarial drug. We propose that shikimate 3-phosphate inhibits the essential *Plasmodium* shikimate biosynthesis pathway without side effects in the host cell (*Figure 6*, *Supplementary file 6*). Excitingly, shikimate 3-phosphate has been used to treat *E. coli* and *Streptococcus* infections without appreciable toxicity for patients (*Díaz-Quiroz et al., 2018*). Furthermore, recent studies have shown that inhibiting the

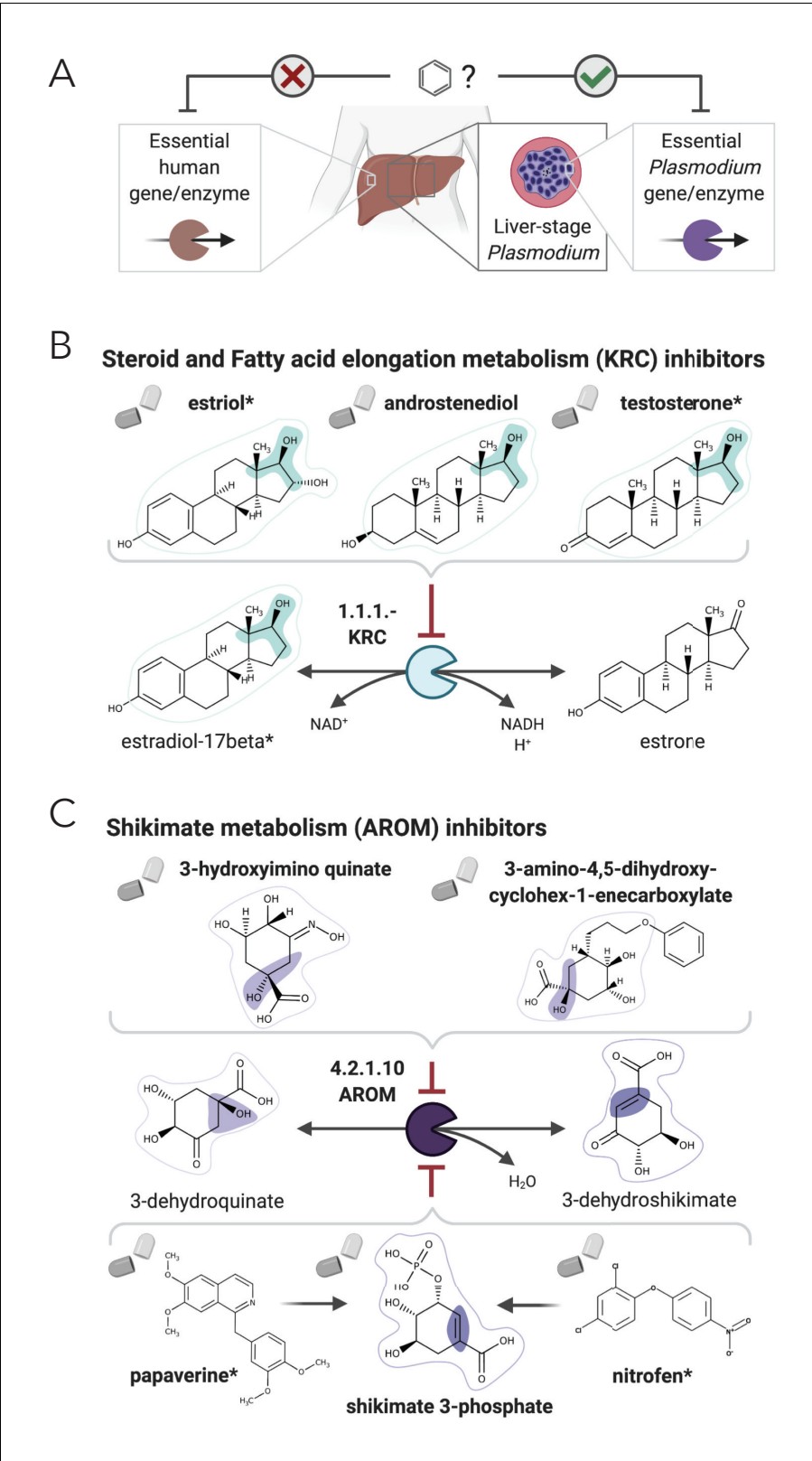

**Figure 6.** NICEdrug.ch suggests shikimate 3-phosphate as a top candidate to target liver-stage malaria and minimize side effects in host human cells. (**A**) Schema of ideal scenario to target malaria, wherein a drug efficiently inhibits an essential enzyme for malaria parasite survival and does not inhibit essential enzymes in the host human cell to prevent side effects. (**B**) Shikimate 3-phosphate inhibits enzymes in the *Plasmodium* shikimate metabolism, *Figure 6 continued on next page*

*Figure 6 continued*

which is essential for liver-stage development of the parasite. Shikimate 3-phosphate does not inhibit any enzyme in the human host cell since it is not a native human metabolite, and it does not show similarity to any native human metabolite. (C) Mechanistic details of inhibition of aroC by shikimate 3-phosphate and other NICEdrug candidates. See also *Supplementary file 5*; *Supplementary file 6*.

shikimate pathway using 7-deoxy-sedoheptulose is an attractive antimicrobial and herbicidal strategy with no cytotoxic effects on mammalian cells (*Brilisauer et al., 2019*). Experimental studies should now validate the capability of shikimate 3-phosphate to efficiently and safely target liver malaria, and could further test other NICEdrug.ch antimalarial candidates (*Supplementary file 6*).

## NICEdrug.ch identifies over 1300 molecules to fight COVID-19, with N-acetylcysteine as a top candidate

SARS-CoV-2 is responsible for the currently on-going COVID-19 pandemic and the death of over three million people (as of today, 11 May 2021 [*Dong et al., 2020*]), and there is currently no confirmed treatment for it. Attacking the host factors that allow replication and spread of the virus is an attractive strategy to treat viral infections like COVID-19. A recent study has identified 332 interactions between SARS-CoV-2 proteins and human proteins, which involve 332 hijacked human proteins or host factors (*Gordon et al., 2020*). Here, we first used NICEdrug.ch to identify inhibitors of enzymatic host factors of SARS-CoV-2. Targeting such human enzymes prevents interactions between human and viral proteins (PPI) ('Materials and methods', *Figure 7A*). Of the 332 hijacked human proteins, we identified 97 enzymes ('Materials and methods', *Supplementary file 7*) and evaluated their druggability by inhibitors among the 250,000 small molecules in NICEdrug.ch and 80,000 molecules in food ('Materials and methods', *Figure 7A*). NICEdrug.ch suggests 22 hijacked human enzymes can be drug targets and proposed 1301 potential competitive inhibitors from the NICEdrug.ch database. Of 1301 potential inhibitors, 465 are known drugs, 712 are active metabolic products of 1419 one-step-away prodrugs, and 402 are molecules in fooDB (*Supplementary file 7*). We found among the top anti SARS-CoV-2 drug candidates the known reverse transcriptase inhibitor didanosine (*Figure 7B*, *Supplementary file 7*), which other in silico screenings have also suggested as a potential treatment for COVID-19 (*Alakwaa, 2020*; *Cava et al., 2020*). Among others, NICEdrug.ch also identified: (1) actodigin, which belongs to the family of cardiotonic molecules proven to be effective against MERS-CoV but without mechanistic knowledge (*Ko et al., 2020*), (2) three molecules in ginger (6-paradol, 10-gingerol, and 6-shogaol) inhibiting catechol methyltransferase, and (3) brivudine, a DNA polymerase inhibitor that has been used to treat herpes zoster (*Wassilew, 2005*) and prevent MERS-CoV infection (*Park et al., 2019*), and NICEdrug.ch suggests it for repurposing (*Figure 7—figure supplement 1*, *Supplementary file 7*).

Drugs like remdesivir, EIDD-2801, favipiravir, and inhibitors of angiotensin converting enzyme 2 (ACE2) have been used to treat COVID-19 (*Jeon et al., 2020*), and act through a presumably effective inhibitory mechanism (*Figure 7—figure supplement 2*). For instance, the three drugs remdesivir, EIDD-2801, and favipiravir are believed to inhibit the DNA-directed RNA polymerase (EC: 2.7.7.6). Here, we used the NICEdrug.ch reactive site-centric fingerprint to search for alternative small molecules in NICEdrug.ch and fooDB that could be repurposed to target ACE2 and DNA-directed RNA polymerase. NICEdrug.ch identified a total of 215 possible competitive inhibitors of ACE2. Among those is captopril, a known ACE2 inhibitor (*Kim et al., 2003*), and D-leucyl-N-(4-carbamimidoylbenzyl)-L-prolinamide, a NICEdrug.ch suggestion for drug repurposing to treat COVID-19. We also found 39 food-based molecules with indole-3-acetyl-proline (a molecule in soybean) as top ACE2 inhibitor candidate (*Figure 7—figure supplement 2*, *Supplementary file 8*). To target the same enzyme as remdesivir, EIDD-2801, and favipiravir, NICEdrug.ch identified 1115 inhibitors of the DNA-directed RNA polymerase, like the drug vidarabine, which shows broad spectrum activity against DNA viruses in cell cultures and significant antiviral activity against infections like the herpes viruses, the vaccinia virus, and varicella zoster virus (*Suzuki et al., 2006*). We further found 556 molecules in food that might inhibit DNA-directed RNA polymerase, like trans-zeatin riboside triphosphate (FDB031217) (*Supplementary file 8*).

One of the host factors identified by *Gordon et al., 2020* is the histone deacetylase 2 (HDAC2), which acetylates proteins and is an important transcriptional and epigenetic regulator. The acetyl

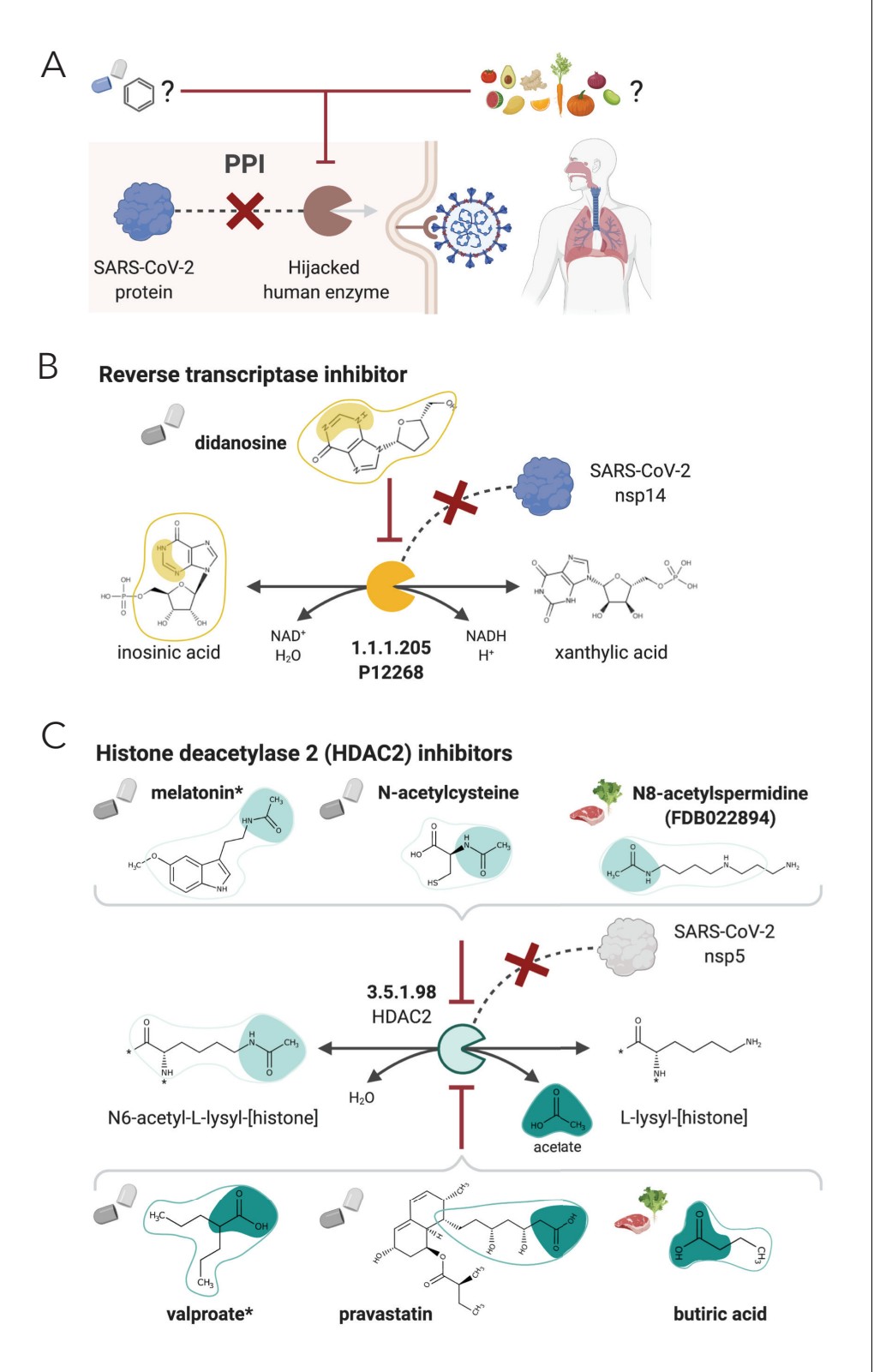

**Figure 7.** NICEdrug.ch strategy to fight COVID-19, and NICEdrug.ch candidate inhibitors of SARS-CoV-2 host factors: reverse transcriptase and HDAC2. (**A**) Schema of NICEdrug strategy to target COVID-19, wherein a drug (top-left) or molecules in food (top-right) efficiently inhibit a human enzyme hijacked by SARS-CoV-2. Inhibition of this host factor reduces or abolishes protein–protein interactions (PPI) with a viral protein and prevents SARS-CoV-2 proliferation. (**B**) Inhibition of the reverse transcriptase (EC: 1.1.1.205 or P12268) and the PPI with SARS-CoV-nsp14 by didanosine based on
*Figure 7 continued on next page*

*Figure 7 continued*

NICEdrug.ch. (**C**) Inhibition of the HDAC2 (EC: 3.5.1.98) and the PPI with SARS-CoV-nsp5 by molecules containing acetyl moiety (like melatonin, N-acetylcysteine, and N8-acetylspermidine), and molecules containing carboxylate moiety (like valproate, stains, and butyrate) based on NICEdrug.ch. See also *Figure 7—figure supplement 1*; *Figure 7—figure supplement 2*, *Supplementary file 7*; *Supplementary file 8*.

The online version of this article includes the following figure supplement(s) for figure 7:

**Figure supplement 1.** NICEdrug candidate inhibitors of SARS-CoV-2 host factors: galactosidase, catechol methyltransferase, and DNA polymerase.
**Figure supplement 2.** NICEdrug candidate inhibitors of ACE2.

and carboxylate moieties are the reactive sites of the forward (N6-acetyl-l-lysyl-[histone]) and reverse (acetate) biotransformation of HDAC2, respectively (*Figure 7*). NICEdrug.ch recognized a total of 640 drugs for repurposing that can inhibit HDAC2, including 311 drugs sharing the acetyl moiety and showing a NICEdrug score above 0.5 with respect to N6-acetyl-L-lysyl-[histone], and 329 drugs sharing the carboxylate moiety and presenting a NICEdrug score above 0.5 with acetate ('Materials and methods'). Among the drugs sharing the acetyl reactive site, we identified the known HDAC2 inhibitor melatonin (*Wu et al., 2018*), and to our knowledge new candidates like N-acetylhistamine and N-acetylcysteine. We also located 22 molecules in food with potential HDAC2 inhibitory activity, like N8-acetylspermidine (FDB022894) (*Figure 7C*, *Supplementary file 8*). Drugs sharing the carboxylate reactive site (as identified with NICEdrug) include the known HDAC2 inhibitors valproate, butyrate, phenyl butyrate (*Abdel-Atty et al., 2014*) and statins (*Kong et al., 2004*; *Figure 7C*, *Supplementary file 8*). Interestingly, statins have been shown to have protective activity against SARS-CoV-2 (*Lodigiani et al., 2020*; *Zhang et al., 2020*). In addition, the NICEdrug.ch candidate N-acetylcysteine is a commonly used mucolytic drug that is sometimes considered as a dietary supplement and has putative antioxidant properties. Indeed, N-acetylcysteine is believed for long to be a precursor of the cellular antioxidant glutathione (*Mårtensson et al., 1989*), but has unknown pharmacological action. NICEdrug.ch suggests that N-acetylcysteine might present a dual antiviral activity: firstly, N-acetylcysteine is converted to cysteine by HDAC2 and by that means, it is competitively inhibiting the native function of HDAC2 and interactions with viral proteins (*Figure 7C*, *Supplementary file 8*). Cysteine next fuels the glutathione biosynthesis pathway and produces glutathione in two steps.

Given the high coverage of validated molecules with activity against SARS-CoV-2 that NICEdrug.ch captured in this unbiased and reactive site-centric analysis, we suggest there might be other molecules in the set of 1300 NICEdrug.ch candidates that could also fight COVID-19. Excitingly, there are many molecules that can be directly tested since these are drugs that have already passed all safety regulations or are molecules present in food, like N-acetylcysteine for which we further reveal an action mechanism behind its potential anti-SARS-CoV-2 activity. Other new candidates for which no safety data is available should be further validated experimentally and clinically. The mechanistic analyses provided by NICEdrug.ch could also guide new pharmacokinetic and pharmacodynamic models simulating SARS-CoV-2 infection and treatment.

## Discussion

To systematically illuminate the metabolism and all enzymatic targets (competitively inhibited) of known drugs and hypothetical prodrugs to aid in the development of new therapeutic compounds, we used a proven reaction–prediction tool BNICE.ch (*Hatzimanikatis et al., 2005*) and an analysis of neighboring atoms of reactive sites analogous to BridgIT (*Hadadi et al., 2019*) and performed the first large-scale computational analysis of drug biochemistry and toxicity in the context of human metabolism. The analysis involved over 250,000 small molecules, and curation and computation of bio- and physico-chemical drug properties that we assembled in an open-source drug database NICEdrug.ch that can generate detailed drug metabolic reports and can be easily accessed and used by researchers, clinicians, and industry partners. NICEdrug.ch revealed 20 million potential reactive sites at the 250,000 small molecules of the database, and there exist over 3000 enzymes in the human metabolism that can be inhibited with the 250,000 molecules. This is because NICEdrug.ch can identify potential metabolic intermediates of a drug and scans these molecules for substructures that can interact with catalytic sites across all enzymes in a desired cell.

NICEdrug.ch adapts the metric previously developed for reactions in BridgIT (*Hadadi et al., 2019*) to precisely compare drug–drug and drug–metabolite pairs based on similarity of reactive site and the neighborhood around this reactive site, which we have recently shown outperforms previously defined molecular comparison metrics (*Hadadi et al., 2019*). Since NICEdrug.ch shows high specificity in the identification of such reactive sites and neighborhod, it provides a better mechanistic understanding than currently available methods (*Robertson, 2005*). Despite these advances, it remains challenging to systematically identify non-competitive inhibition or targeting of non-enzymatic biological processes. We suggest coupling NICEdrug.ch drug metabolic reports with other in silico and experimental analyses accounting for signaling induction of small molecules and other non-enzymatic biological processes like transport of metabolites in a cell. The combined analysis of drug effects on different possible biological targets (not uniquely enzymes) will ultimately increase the coverage of molecules for which a mechanistic understanding of their mode of action is assigned.

A better understanding of the mechanisms of interactions and the specific nodes where the compounds act can help re-evaluate pharmacokinetic and pharmacodynamic models, dosage, and treatment. Such understanding can be used in the future to build models that correlate the pharmacodynamic information with specific compounds and chemical substructures in a manner similar to the one used for correlating compound structures with transcriptomic responses. We have shown for one of the most commonly used anticancer drugs, 5-FU, that NICEdrug.ch identifies and ranks alternative sources of toxicity and hence can guide the design of updated models and treatments to alleviate the drug's side effects.

The mechanistic understanding will also further promote the development of drugs for repurposing. While current efforts in repurposing capitalize on the accepted status of known drugs, some of the issues with side effects and unknown interactions limit their development as drugs for new diseases. Given that drug repurposing will require new dosage and administration protocols, the understanding of their interactions with the human metabolism will be very important in identifying, developing, and interpreting unanticipated side effects and physiological responses. We evaluated the possibility of drug repurposing with NICEdrug.ch as a substitute for statins, which are broadly used to reduce cholesterol but have many side effects. NICEdrug.ch and its reactive site-centric comparison accurately cluster both family types of statins, even though they are similar in overall molecular structure and show different reactivity. In addition, NICEdrug.ch suggests a set of new molecules with hypothetically less side effects (*Endo and Hasumi, 1993*; *Tanaka et al., 2004*) that share reactive sites with statins.

A better mechanistic understanding of drug targets can guide the design of treatments against infectious diseases, for which we need effective drugs that target pathogens without side effects in the host cell. This is arguably the most challenging type of problem in drug design, and indeed machine learning has continuously failed to guide such designs given the difficulty in quantifying side effects – not to mention in acquiring large, consistent, and high-quality data sets from human patients. To demonstrate the power of NICEdrug.ch for tackling this problem, we sought to identify drugs that target liver-stage malaria parasites and minimize the impact on the human host cell. We identified over 500 drugs that inhibit essential *Plasmodium* enzymes in the liver stages and minimize the impact on the human host cell. Our top drug candidate is shikimate 3-phosphate targeting the parasite's shikimate metabolism, which we recently identified as essential in a high-throughput gene knockout screening in *Plasmodium* (*Stanway et al., 2019*). Excitingly, our suggested antimalarial candidate shikimate 3-phosphate has already been used for *Escherichia* and *Streptococcus* infections without appreciable side effects (*Díaz-Quiroz et al., 2018*).

Finally, minimizing side effects becomes especially challenging in the treatment of viral infections, since viruses fully rely on the host cell to replicate. As a last demonstration of the potential of NICEdrug.ch, we sought to target COVID-19 by identifying inhibitors of 22 known enzymatic host factors of SARS-CoV-2 (*Gordon et al., 2020*). NICEdrug.ch identified over 1300 molecules that might target the 22 host factors and prevent SARS-CoV-2 replication. As a validation, NICEdrug.ch correctly identified known inhibitors of those enzymes and further suggested safe drugs for repurposing and other food molecules with activity against SARS-CoV-2. Among the NICEdrug.ch suggestions for COVID-19, based on the knowledge on its mechanism and safety, we highlight N-acetylcysteine as an inhibitor of HDAC2 and SARS-CoV-2.

Overall, we believe that a system-level or metabolic network analysis, coupled with an investigation of reactive sites, will likely accelerate the discovery of new drugs and provide additional understanding regarding metabolic fate, action mechanisms, and side effects and can complement ongoing experimental effects to understand drug metabolism (*Javdan et al., 2020*). To fully capture, understand, and predict drug metabolism, it is necessary to evaluate two aspects: (1) the metabolic fate of small molecules and (2) the absorption and distribution of small molecules to the actual target cells and enzymes. This study concentrates on the first aspect, whereas the second aspect will be addressed in future work on NICEdrug.ch.

We suggest the generation of drug metabolic reports to understand the reactivity of new small molecules, the possibility of drug repurposing, and the druggability of enzymes. Our results and high predictive accuracy (above 70%) using NICEdrug.ch suggest that this database can be a novel avenue towards the systematic pre-screening and identification of drugs and antimicrobials. In addition to human metabolic information, NICEdrug.ch currently includes information for the metabolism of *P. berghei* and *E. coli*. Because we are making it publicly available (https://lcsb-databases.epfl.ch/pathways/Nicedrug/), our hope is that scientists and decision takers in pharmaceutical industry alike can make use of this unique database to better inform their research and clinical decisions – saving time, money, and ultimately lives.

# Materials and methods

**Key resources table**

| Reagent type (species) or resource | Designation | Source or reference | Identifiers | Additional information |
| --- | --- | --- | --- | --- |
| Software, algorithm | OpenBabel 2.4.1 | doi:10.1186/1758-2946-3-33 | | |
| Software, algorithm | BridgIT | doi:10.1073/pnas.1818877116 | | |
| Software, algorithm | ATLAS of biochemistry | doi:10.1021/acssynbio.6b00054; doi:10.1021/acssynbio.0c00052 | | |
| Software, algorithm | MORPHEUS | https://clue.io/morpheus | | |
| Software, algorithm | NICEdrug.ch (curated bioactive molecules and analysis of drug metabolism) | This paper; http://nicedrug.ch/ | | See Materials and methods |

## Representation of metabolic neighborhood in figures of this manuscript

We represent the metabolic neighborhood of a drug with reactions or steps away (arrows), where each step away (circle connected to arrow) involves a set of compounds. We extract compounds at each step that present a high NICEdrug score (value under metabolite name) with the native substrate of a reaction in the human cell. Reactive sites common to neighbor metabolites and native human metabolites are shaded with colors matching the color of the enzymes (packmen) that are inhibited. The neighborhood (seven atoms away, as considered in NICEdrug score) of the reactive sites is circled in the metabolites and native human metabolites with the same color as the reactive sites and enzymes. Compounds marked with * are confirmed inhibitors and references are provided in the main text.

## Representation of enzymatic inhibition in figures of this manuscript

We represent the enzymes and catalyzed reactions inhibited by NICEdrug candidates. Highlighted are the reactive site and neighborhood (as considered in the NICEdrug score) in candidate drugs and metabolites, which are native substrates of the human enzymes. The SARS-CoV-2 proteins interaction with the enzyme is presumed to be diminished or abolished upon inhibition of the human enzyme. Compounds marked with * are confirmed inhibitors and references are provided in the main text.

## Curation of input molecules used in the construction of NICEdrug.ch

We constructed the NICEdrug.ch database to gather small molecules suitable for treatment of human diseases. We collected the SMILES structure, synonyms, and any available bio- and physico-chemical property included from three source databases: KEGG, ChEMBL, and DrugBank, which added up to 70,976 molecules by January 2018 (*Figure 1—figure supplement 1, A*). Only molecules that were fully structured were imported to our database. We further curated the imported molecules by removing duplicate structures and merging annotations from different databases into one molecule entry in the database. For removing duplicate structures we used canonical SMIELS (*Weininger, 1988*) generated by openbabel (*O'Boyle et al., 2011*) version 2.4.0. This unification method is based on atoms and their connectivity in a molecule in terms of a molecular graph that is captured by the canonical SMILES. Therefore, different resonance forms, stereoisomers, as well as dissociated and charged states of the same compound are mapped to one entry in database. Furthermore, we filtered all molecules collected from the ChEMBL database based on Lipinski rules in an attempt to discard chemicals less likely to show drug properties. The Lipinski rules are as follows (*Lipinski et al., 2001*): (1) the molecular weight should be less than 500 Dalton, (2) the number of hydrogen bond donors should be less than five, (3) the number of hydrogen bond acceptors should be less than 10, and (4) an octanol-water partition coefficient (log P) should be less than five. According to Lipinski rules, an active oral drug does not violate more than one of the above criteria. We calculated criteria one, two. and three based on the structural information from SMILES of molecules. To assess criterion four, we relied on reported data in the source database. We kept in the NICEdrug.ch database all those compounds for which the partition coefficient was not available.

We performed a separate analysis to account for non-unique graph representations of aromatic rings, also called *kekulé structures*. The existence of aromatic rings and the fact that bond-electrons are shared within the ring make several single-double bond assignments possible, which results in multiple kekulé representations for a single molecule (*Figure 1—figure supplement 1, B*). We included all such kekulé structures to account for alternative atom-bond connectivity and associated reactivity. We call 'effective forms' to the kekulé representations that show different reactive sites than their canonical structures. For example, there can be two effective forms plus the canonical structure (*Figure 1—figure supplement 1, B*). In total, we found 42,092 effective forms for 29,994 aromatic compounds in NICEdrug.ch database and we kept them for further analysis.

We also computed the thermodynamic properties of all drugs in NICEdruch.ch. Specifically, we computed the Gibbs free energy of formation ($_fG^o$) using the group contribution method of Mavrovouniotis (*Jankowski et al., 2008*).

The NICEdrug.ch database includes a total number of 48,544 unique and curated small molecules (*Figure 1—figure supplement 1A*).

## Identification of reactive sites in drugs and drug candidates

The 3D structures of enzyme pockets are complex and mostly unknown. Therefore, evaluating and comparing docking of two small molecules in the pocket of a specific target is impossible most of the times. Using BNICE.ch, we focused on the complementary structure of active sites on substrates, also called *reactive site*. To recognize the potential reactive sites on molecules, we scanned molecules using expert-curated generalized reaction rules of BNCIE.ch (*Hadadi et al., 2016*), which mimic the identification of substrates by the enzyme pocket and account for the promiscuous activity of enzymes. Theses reaction rules incorporate the information of biochemical reactions and have third-level Enzyme Commission (EC) identifiers. Each BNICE.ch reaction rule accounts for three levels of information: (1) atoms in reactive sites of compounds, (2) connectivity and configuration of atom bonds in the reactive site, and (3) mechanism of bond breakage and formation during the reaction. As of May 2020, BNICE.ch contains 450 bidirectional generalized reaction rules that can reconstruct 8118 KEGG reactions (*Hadadi et al., 2016*). Here, we include all BNICE.ch rules to identify all possible reactive sites on a given molecule in two steps. First, a BNICE.ch rule identifies all atoms in a compound that belong to the rule's reactive site. Second, the rule evaluates the connectivity of the atoms previously identified. The candidate compounds for which a BNICE.ch rule identified a reactive site were validated as metabolically reactive and considered for analysis in NICEdrug.ch.

It is important to note that thanks to the generalized reaction rules, which abstract the knowledge of thousands of biochemical reactions, BNICE.ch is able to reconstruct known biotransformations

and also propose novel metabolic reactions. This was demonstrated in the reconstruction of the ATLAS of Biochemistry (*Hadadi et al., 2016*), which involves up to 130,000 reactions between known compounds.

## Analysis of drug metabolism in human cells

To mimic biochemistry of human cells and simulate human drug metabolism, we collected all available information (metabolites and metabolic activities or EC numbers of enzymes) on human metabolism from three available databases: the human metabolic models Recon3D (*Brunk et al., 2018*) and HMR (*Pornputtapong et al., 2015*), and the Reactome database (*Croft et al., 2011*). These three databases include a total of 2266 unique human metabolites and 2066 unique EC numbers of enzymes (*Supplementary file 1*).

To explore the biochemical space beyond the known human metabolic reactions and compounds, we used (1) the generalized enzymatic reaction rules of BNICE.ch that match up to the third EC level the collected human enzymes and (2) all of the collected human metabolome. We evaluated the reactivity of each drug or drug candidate in a human cell using the retro-biosynthesis algorithm of BNICE.ch, which predicts hypothetical biochemical transformations or *metabolic neighborhood* around the drug candidate of study. We generated with BNICE.ch metabolic reactions in which each drug candidate and all known human metabolites could participate as substrate or products. We also allowed a set of 53 known cofactors to react with the human metabolites (*Supplementary file 1*).

We define the boundaries of the metabolic neighborhood of a molecule with a maximum number of reactions or *steps away* that separate the input molecule (drug candidate of study) from the furthest compound. In BNICE.ch, a generation *n* of compounds involves all metabolites that appear for the first time in the metabolic neighborhood of a drug candidate after *n* reactions or steps happened. For example, in the case study of 5-FU, we find the compound 5-fluorouridine in generation 2 or two steps away, which means that there are two metabolic reactions that separate 5-FU and 5-fluorouridine (*Figure 2*).

In NICEdrug.ch, there exist 197,246 compounds in generation 1 (one step away) from all input drug candidates. The 197,246 compounds are part of the potential drug metabolic neighborhood in human cells. Out of all generation 1 molecules, 13,408 metabolites can be found in human metabolic models and HMDB database (*Wishart et al., 2018*), 16,563 metabolites exist in other biological databases, and the remaining 167,245 metabolites are catalogued as known compounds in chemical databases (i.e., PubChem). Note that HMDB includes native human metabolites and non-native human compounds, like food ingredients.

The 197,246 products that are onestep away of all NICEdrug.ch molecules are part of a hypothetical biochemical neighborhood of 630,449 drug metabolic reactions. Of all drug metabolic reactions, 5306 reactions are cataloged in biological databases, and the remaining 625,143 reactions are novel. A majority of the reactions involved oxidoreductases (42.54%), broken down into 27.45% of lyases, 7.15% of hydrolases, 6.28% of transferases, 1% of isomerases, and 15.58% of ligases. Interestingly, based on the previously identified reactive sites, of the 265,935 (42.54% of 625,143) oxidoreductase reactions, 49.92% are catalyzed by the p450 family of enzymes, which are known to be responsible for the metabolism of drug (*Figure 1—figure supplement 2C*).

## Using NICEdrug.ch database for analysis of the metabolic neighborhood of a drug candidate

In NICEdrug.ch webserver, users can look up for a drug using the drugs' name and other identifiers like ChEMBL, DrugBank, and KEGG. NICEdrug.ch will report a unique identifier for the compound that will be input for upcoming analysis modules. The *reactivity* module allows to study the metabolic network around an input molecule. The input to this module is as follows: (1) the unique identifier of the drug of interest and (2) a maximum number of reactions or *steps away* that shall separate the input drug to the furthest compound in the metabolic neighborhood.

The output of this analysis is a report in the form of a csv file that includes all compounds and metabolic reactions in the metabolic neighborhood of the input drug. One can also export the neighborhood in the form of a visual graph, in which nodes are molecules and edges are reactions.

## Definition of the NICEdrug score

Based on the theory of lock and key, two metabolites that can be catalyzed by the same enzyme may have similar reactive sites and also neighboring atoms. In order to quantify the similarity inside and around reactive sites of <u>two molecules</u>, we developed a metric called *NICEdrug score* (*Figure 1—figure supplement 3*), which is inspired on BridgIT (*Hadadi et al., 2019*). BridgIT assesses the similarity of <u>two reactions</u>, considering the reactive site of the participating substrates and their surrounding structure until the seventh atom out of the reactive site.

The NICEdrug score is an average of two similarity evaluations: (1) the atom-bond configuration inside reactive site ($\alpha$ parameter) and (2) the seven atom-bond chain molecular structure around the reactive site ($\beta$ parameter). The NICEdrug score, and its parameters $\alpha$ and $\beta$, range between 0 and 1 when they indicate no similarity and identical structure, respectively. Different constraints on the $\alpha$ and $\beta$ parameters determine the identification of different types of inhibition like para-metabolites and anti-metabolites (see other sections in this 'Materials and methods').

We show the evaluation of NICEdrug scores for three example compounds (*Figure 1—figure supplement 3*). In this example, Digoxin, Labriformidin, and Lanatoside C all share the reactive site corresponding to EC number 5.3.3.- ($\alpha$=1). Starting from the atoms of the identified reactive site, eight description layers of the molecule were formed, where each layer contains a set of connected atom-bond chains. Layer zero includes types of atoms of reactive site and their count. Layer one expands one bond away from all of the atoms of reactive site and accounts for atom-bond-atom connections. This procedure is continued until layer 7, which includes the sequence of eight atoms connected by seven bonds. Then, we compare the fingerprint of each molecule to the other participants of the class based on the Tanimoto similarity scores. A Tanimoto score near 0 designates no or low similarity, whereas a score near one designates high similarity in and around reactive site. Lanatoside C and Digoxin share the same substructure till eight layers out of reactive site which is presented in the NICEdrug score by preserving score one in all layers, so the overall Tanimoto score for these two compounds in the context of EC number 5.3.3.- is 1 ($\alpha$=1 and $\beta$=1). However, the structure of two compounds are not exactly the same, and actually Lanatoside C has eight more carbon atoms and six more oxygen atoms, shaped as an extra benzenehexol ring and an ester group. Although this part is far from the reactive site, based on the NICEdrug score, they both can perfectly fit inside the binding pocket of a common protein related to this reactive site. This hypothesis is proved by experiments reported in KEGG and DrugBank. According to DrugBank and KEGG, Lanatoside C has actions similar to dioxin and both of them have the same target pathways: cardiac muscle contraction and adrenergic signaling in cardiomyocytes. Furthermore, target protein for both of them is ATP1A.

Also, the NICEdrug score effectively captures and quantifies differences around the reactive site. The substructure around the reactive site in Labriformidin is slightly different ($\alpha$=1 and $\beta$ <1). The difference is calculated trough different layers of the NICEdrug score.

In the case study of 5-FU, in order to predict competitive inhibition, we analyzed all the metabolites that share reactive site with 5-FU or its downstream products ($\alpha$=1) and then we ranked the most similar metabolites based on their similarity in neighborhood of reactive site to 5-FU or its downstream products ($\beta$). To assess the structural differences in the reactive sites themselves ($\alpha$), we implemented the Levenshtein edit distance algorithm (*Levenshtein, 1966*) to determine how many deletions, insertions, or substitutions of atom/bonds are required to transform one pattern of reactive site into the other one. Here, the edit distance explains the difference between the reactive sites of the intermediate and the human metabolite. However, even slight changes in the reactive site affect its interaction with the binding site. To ensure that the divergence retained the appropriate topology, we compared the required edit on reactive site with interchangeable groups, termed bioisosteric groups (*Papadatos and Brown, 2013*). These bioisosteric groups contain similar physical or chemical properties to the original group and largely maintain the biological activity of the original molecule. An example of this is the replacement of a hydrogen atom with fluorine, which is a similar size that does not affect the overall topology of the molecule. For this analysis, we used 12 bioisosteric groups adapted from the study by *Papadatos and Brown, 2013*.

To predict irreversible Inhibitors in metabolism of 5-FU, we kept only molecules with a similarity score greater than 0.9 to metabolites ($\beta$>0.9), to preserve a high similarity in the neighborhood of

the reactive sites. Then, we checked which ones contained reactive sites that differed only in the replacement of bioisosteric groups ($\alpha\sim1$).

## How to interpret the NICEdrug score

The NICEdrug score defines the similarity of two reactive site-centric fingerprints. If two molecules show a high NICEdrug score, there is a high probability that they will both fit inside the enzymatic pocket of the same protein. In a previous study, we introduced the reaction reactive site-centric fingerprint to compare all ligands of a reaction simultaneously, also called BridgIT fingerprint (*Hadadi et al., 2019*). We compared reactions based on their BridgIT fingerprint and their catalyzing enzymes based the homology of the protein sequence.

We then compared the similarity between catalyzing enzymes with the similarity of BridgIT fingerprints for their corresponding reactions. This comparison served to assess how accurately the BridgIT fingerprint predicts protein sequence similarity and putative catalyzing proteins for a reaction. A high similarity between protein sequences and BridgIT fringerprints represented a true positive. We found that with a threshold of 0.3 and 0.5 for the BridgIT fingerprint we reached an accuracy of 0.8 and 0.85, respectively, being the E-value from the BLAST of the protein sequences $10^{-10}$ and $10^{-50}$ (*Hadadi et al., 2019*).

This analysis guided the definition of the NICEdrug score based on the reactive site-centric fingerprint. In this manuscript, we defined the threshold of 0.3 for the NICEdrug score to assign similarity in reactivity between two molecules and of 0.5 to identify enzyme targets for a small molecule.

Moreover, the threshold on the NICEdrug score is consistent with the results of large-scale druggability analysis ('Materials and methods', evaluation of the NICEdrug.ch druggability report), where we identified an optimal similarity threshold of 0.46 with values of accuracy, precision, recall, and F1 of 0.84, 0.88, 0.89, and 0.89, respectively.

## Classification of drugs based on the NICEdrug score

Classification of compounds with similar structure is normally used to assign unknown properties to new compounds. For instance, one can infer ligand-protein binding for a drug when its action mechanism or the structure of the target proteins are not known. In this study, we have demonstrated four strategies to classify drugs (*Figure 1*), which are from less to more stringent: classifying (1) molecules that participate in reactions with the same EC up to the third level, (2) molecules that in addition share a BNICE.ch reaction rule, (3) molecules that in addition to both previous points share reactive site, and (4) molecules that show high similarity of reactive site and neighborhood based on the NICEdrug score.

The EC number guarantees that molecules are catalyzed with similar overall reaction mechanism. Generalized reaction rules from BNICE.ch further capture different submechanisms inside an EC number (*Hadadi et al., 2016*). A BNICE.ch reaction rule might involve more than one reactive site. Hence, information of reactive sites provide further insights into the molecule's reactivity. Furthermore, similarity of reactive sites and their neighborhoods based on the NICEdrug score increase the comparison resolution and this is the basis of the classification in NICEdrug.ch.

In NICEdrug.ch database, there exist 95,342 classes that comprise all drugs and human compounds sharing EC, BNICE.ch rule, and reactive site (classification based on our strategy 3). We computed the NICEdrug score between all pairs of molecules in a class and this information is available in NICEdrug.ch.

## Identification of drugs acting as para-metabolites based on NICEdrug score

Small molecules that share reactive site and are structurally similar to native human metabolites enter and bind the pocket of native enzymes and competitively inhibiting catalysis acting as para-metabolites (*Ariens, 2012*). In this study, we define para-metabolite as any drug or any of its metabolic neighbors that (1) shares reactive site with native metabolites ($\alpha=1$) and (2) preserves a high NICEdrug score with respect to the reactive site neighborhood ($\beta>0.9$).

## Identification of drugs acting as anti-metabolites based on NICEdrug score

Small molecules that do not share reactive site but are structurally similar to native human metabolites might enter the binding pocket of native enzymes and inhibiting catalysis acting as anti-metabolites (*Ariens, 2012*). In this study, we define anti-metabolite as any drug or any of its metabolic neighbors that (1) differs slightly in reactive site from a native metabolite (α~1) and (2) preserves high similarity in the reactive site neighborhood (β>0.9). We hypothesize that a low divergence in the reactive site, still allows a non-native compound to enter and bind the enzyme pocket since it is structurally similar enough to the native substrate.

## Identification of NICEdrug toxic alerts

We obtained all NICEdrug toxic alters from ToxAlert database (*Sushko et al., 2012*). ToxAlert database includes about 1200 structural toxic alerts associated with particular types of toxicity. Toxic alerts are provided in the form of SMART patterns that are searchable in SMILES structure of molecules. NICEdrug.ch uses openbable tool (*O'Boyle et al., 2011*) to search for these structural alerts on SMILES of compounds.

## Collection of reference toxic molecules in NICEdrug.ch

Studying the adverse effects of chemicals on biological systems has led to development of databases cataloging toxic molecules. The Liver Toxicity Knowledge Base (LTKB) integrates 1036 molecules annotated with human Drug-induced liver injury risk (severity). Super toxic DB include about 60 k toxic molecules, which are annotated with their toxicity estimate, $LC_{50}/LD_{50}$ i.e., lethal dose or concentration at which 50% of a population dies.

As a resource of approved toxic molecules, we collected all of the molecules cataloged as toxic in LTKB and super toxic databases. We used this collection as a reference to compare the similarity of drugs or/and products of drug metabolism with approved toxic molecules.

## Definition of a toxicity score in NICEdrug.ch

The number of molecules labeled as toxic in databases is disproportionally low compared to the space of compounds. On the other hand, toxic alerts are defined for a big number of compounds and are linked to redundant molecular structures.

We measured the similarity of drugs and their metabolic neighbors with the collection of reference toxic molecules using the NICEdrug score. We assigned toxic alerts to molecules in NICEdrug. ch if a molecule and toxic molecule shared a molecular substructure linked to the toxic alert.

Finally, NICEdrug.ch provides a toxicity report in the form of a csv file for each molecule in the metabolic neighborhood including six values linked to the most similar toxic molecules in both toxic reference databases (LTKB and supertoxic databases): (1) the NICEdrug score between the drug and those most similar toxic molecules, (2) the severity degree of the hepatotoxic compound, and log ($LC_{50}$) of the supertoxic compound, and (3) the number of common toxic alerts between the drug and the most similar toxic molecules. The list of toxic alerts is also provided.

We combined the six values of the toxicity report into a toxicity score defined as follows:

$$\sum_i \text{NICEdrug score} \times (\log(LC_{50}) \text{ or severity degree}) \times \text{number of common NICEdrug toxic alerts}$$

$$i \in \{\text{the most similar approved toxic molecules in LTKB and supertoxic databases}\}$$

The toxicity score in NICEdrug.ch served to quantify the toxicity of each molecule in the metabolic neighborhood of a drug, recapitulate known toxic molecules, and suggest new toxic compounds (*Figure 4*).

## Analysis of essential enzymes and linked metabolites in *Plasmodium* and human cells

We extracted information of essential genes and enzymes for liver-stage malaria development from our recent study (*Stanway et al., 2019*). In this study, we developed the genome-scale metabolic

model of *Plasmodium berghei*, which shows high consistency (approximately 80%) with the largest gene knockout datasets in *Plasmodium* blood (*Bushell et al., 2017*) and liver stages (*Stanway et al., 2019*). There are 178 essential genes for *P. berghei*'s growth simulating liver-stage conditions (*Stanway et al., 2019*). Here, we identified the substrates of those essential metabolic enzymes, which comprise a set of 328 metabolites (*Supplementary file 5*). To further minimize on the host cell, we filtered out those *Plasmodium* enzymes that share fourth-level EC with human essential enzymes. We used available CRISPR gene essentiality data in various human cell lines (*Wang et al., 2015*) to identify essential genes and enzymes in human cells (*Supplementary file 5*). We further identified essential metabolites in human cells (*Supplementary file 5*) using the latest human genome-scale metabolic model (*Robinson et al., 2020*) and the metabolic information associated to the essential human genes. Subtracting essential parasite and human enzymes resulted in the analysis of 32 essential *Plasmodium* enzymes catalyzing 68 metabolites and 157 unique metabolite-enzyme pairs in the parasite (*Supplementary file 6*).

## Identification of drugs to target malaria and minimize side effects on human cells

Those molecules that themselves and their downstream products cannot act as inhibitors of essential metabolic enzymes in the human host cell, while they can target essential *Plasmodium* enzymes are attractive antimalarial candidates.

We first used NICEdrug.ch to look for small molecules that share reactive site with the 32 essential *Plasmodium* enzymes and they have good similarity score in reactive site neighborhood to native substrates of essential enzymes of parasite, i.e. NICEdrug score above 0.5 (*Supplementary file 6*). We also identified prodrugs that might lead to downstream products with similar reactive site and neighborhood (NICEdrug above 0.5) to any of the essential *Plasmodium* metabolites (*Supplementary file 6*). We suggest those drugs and downstream products act as anti-metabolites and competitively inhibit the essential enzymes in the parasite. Overall, we identified 516 drugs that directly compete with essential metabolites and 1164 prodrugs that need to be biochemically modified between one to three times in human cell to render inhibition of essential enzymes.

We next combined information of essential *Plasmodium* and human metabolites to screen further the drug search using NICEdrug.ch. Of the hypothetical 516 antimalarial candidates, we identified 64 drugs that share reactive site with parasite metabolites (NICEdrug score above 0.5) and not with human metabolites (NICEdrug score below 0.5), making them good candidates for drug design (*Supplementary file 6*).

## Prediction of inhibitors among food-based molecules

We used the reactive site-centric fingerprint available in NICEdrug.ch to identify molecules in food that share reactive site with native substrates of human enzymes and hence might inhibit those enzymes. We retrieved the total set of 80,000 compounds from FooDB (*Scalbert et al., 2011*) and treated them as input molecules into the NICEdrug pipeline (*Figure 1*) to identify reactive sites and evaluate their biochemistry, as done for all molecules in NICEdrug.ch.

## Identification of small molecules to target COVID-19

A recent study reported 332 host factors of SARS-CoV-2 (*Gordon et al., 2020*). Of the 332 proteins, 97 have catalytic function and EC number assigned, and are potential targets of small molecules. We evaluated the druggability of these 97 enzymes using NICEdrug.ch.

To generate a druggability report, NICEdrug.ch first gathers the metabolic reactions associated with the protein EC numbers. NICEdrug.ch uses 11 databases (including HMR, MetaCyc, KEGG, MetaNetX, Reactome, Rhea, Model SEED, BKMS, BiGG models, and Brenda) as source of metabolic reactions. All these databases involve a total of 60 k unique metabolic reactions.

Of the 97 host factor enzymes, we identified 22 enzymes that are linked to fully defined metabolic reactions. Fully defined metabolic reactions fulfill three criteria. (1) There is a secondary structure available for all the reaction participants, which means there are available mol files. (2) There is a fully defined molecular structure for all the reaction participants, which means molecules with unspecified R chains are discarded. (3) There is a BNICE.ch enzymatic reaction rule assigned to the reaction (*Supplementary file 7*).

NICEdrug.ch identified 22 host factor enzymes with 24 unique linked EC numbers and 145 unique fully defined reactions. NICEdrug.ch extracts the metabolites participating in these reactions and identifies their reactive site for a reactive-site centric similarity evaluation against a list of molecules. To this end, NICEdrug.ch reports the list of molecules ranked based on the NICEdrug score. The molecule with the highest NICEdrug score shares the highest reactive site-centric similarity with the native substrate of the target enzyme (*Supplementary file 7*).

We found 1301 molecules that show NICEdrug score above 0.5 with respect to substrates of the 22 SARS-CoV-2 hijacked enzymes (*Supplementary file 7*). Of 1301 molecules, 465 are drugs cataloged in DrugBank, KEGG drugs, or ChEMBL databases; 712 are active molecules one step away of 1419 prodrugs; and 402 are food molecules (*Supplementary file 7*).

To better understand the classes of drugs or food molecules, we classified drugs based on their KEGG drug groups (Dgroups) and food molecules based on their food source. Of 465 drugs identified, 43 drugs are assigned to 55 different Dgroups and 402 food molecules belong to 74 different food sources (*Supplementary file 7*).

## Comparison of NICEdrug.ch predictions and biochemical assays

PubChem's BioAssay (*Wang et al., 2012*) is an open-access database that stores the results of high-throughput screening of diverse set of compounds against different target proteins from the literature (*Wang et al., 2012*). For example, it describes the effect of a drug to target or inhibit one enzyme by defining a drug–enzyme pair as *active* or *inactive*. This collection of data provides an opportunity to investigate the predictive power of NICEdrug.ch.

For each drug in NICEdrug.ch, we collected all evidences about its activity in PubChem's BioAssay database. In a preprocessing step, we removed interactions with non-enzymatic targets (lacking an annotated EC number), multifunctional enzymes (a target enzyme with more than one EC number), and translocase enzymes (new EC class 7, which is out of the scope of NICEdrug.ch).

The final set of molecules and enzymes in NICEdrug.ch for which there is bioassay data available is 2570 small molecules and 198 enzymes. These drugs and enzymes result in over 90 k drug–enzyme pairs, along with their bioassay results (active or inactive). Next, we evaluated the druggability of the 198 enzymes with the 2570 drugs using NICEdrug.ch. We assumed an enzyme is druggable when the NICEdrug.ch score between the drug and the enzyme's native substrate is above 0.3. The NICEdrug.ch ID of the drug and the third-level EC number of the enzyme are the input parameters for this test, which can be reproduced in the NICEdrug.ch server using the druggability report. The third-level of the EC number (instead of the full four digits or fourth-level EC) are commonly used as a description of the biochemistry of an enzyme irrespective to the specific substrates (*Almonacid et al., 2010*; *George et al., 2004*; *Gherardini et al., 2007*; *Hegyi and Gerstein, 1999*; *Todd et al., 2001*). Therefore, two enzymes with the same third level of EC number are expected to catalyze similar biotransformations. In this test, a hit is defined true positive if the predicted drug–enzyme (result of the druggability report of NICEdrug.ch) is described as *active* in the PubChem bioassay result.

The analysis of 2570 small molecules and 198 enzymes with NICEdrug.ch shows the predictions and bioassay results are correlated with an accuracy of 0.73, specificity 0.84, and precision 0.24 (*Supplementary file 2*). Interestingly, a closer inspection shows that the result for 1269 drugs match with 100% accuracy the NICEdrug.ch predictions (true positive hits = 237, true negative hits = 22,764). For the remaining set of 1301 drugs, NICEdrug.ch predictions correlate with the bioassays with an accuracy of 0.65. We hypothesize that the reduced correlation of this set is due to the fact that these molecules do not act as competitive inhibitors. For example, dopamin affects the activity of the enzyme adenylate cyclase through signaling (*Bibb, 2005*).

To have a better understanding of the potential mismatches in the NICEdrug.ch-bioassay comparisons, we investigated the prediction performance over five benchmarked drugs: Tolcapone, Pravastatin, Oxfenicine, Dopamine, and Acarbose. The set of selected drugs cover diverse structural complexity, ranging from small (such as dopamine) to more complex molecules (such as acarbose) with different functionalities. We observed that false predictions in this set are related to (1) not fully characterized enzymes, e.g. multistep or orphan reactions; (2) large diversity in the third level of the EC number of these enzymes; (3) non-metabolic effect, e.g., signaling effect, of these drugs; and (4) contrary bioassay results (*Supplementary file 2*).

## Qualitative comparison of NICEdrug.ch with other metabolic prediction tools

We aimed to compare NICEdrug.ch and other available tools that predict drug metabolism (*Supplementary file 2*). These tools use rule-based or machine learning algorithms to (1) identify the reactive site, also known as site of metabolism, on the small molecule, and/or (2) predict the metabolic products of a small molecule through the generation of a reaction (*Djoumbou-Feunang et al., 2019*).

Computational tools, such as ADME WORKS Predictor (from https://www.fujitsu.com/jp/group/kyushu/en/solutions/industry/lifescience/admeworks/predictor/), ADMET Predictor (Metabolism module from https://www.simulations-plus.com/software/admetpredictor/), admetSAR (*Cheng et al., 2012a*), PreADMET (from https://preadmet.bmdrc.kr/), SMARTCyp (*Rydberg et al., 2010*), SOMP (way2Drug) (*Rudik et al., 2015*), MetaSite (*Cruciani et al., 2005*), RS-WebPredictor (*Zaretzki et al., 2013*), and IDSite (*Li et al., 2011*), are specific to phase I of metabolism and the biochemistry of CYP450 enzymes.

Other tools like Meteor Nexus (*Marchant et al., 2008*), SyGMa (*Ridder and Wagener, 2008*), BioTransformer (*Djoumbou-Feunang et al., 2019*), Meta-PC (*Klopman et al., 1997*), TIMES (*Mekenyan et al., 2004*), and MetaPath (*Handorf and Ebenhöh, 2007*) are more general and cover both phase I and phase II of metabolism. Among the tools covering phase II of metabolism, Meteor Nexus, Meta-PC, and TIMES are only commercially available. Moreover, these tools are mostly focused on mammalian metabolism (e.g. Meteor Nexus).

All available tools present various limitations. In particular, these methods (1) evaluate a small and specific class of metabolic reactions; (2) account only for phase I or phase II of metabolism; and (3) consider uniquely one reaction step to identify reactive sites or metabolic products. In addition, (4) often the mechanism of reaction is not precisely described; and (5) reactions are not enzymatically characterized. Moreover, (6) most of the tools are not freely distributed.

These limitations call for a broader set of computational tools that allow to study and predict the metabolic fate, competitive inhibition, and side effects of small molecules in the context of cellular metabolism like human cells, microbes, parasites, etc.

The NICEdrug.ch resource capitalizes on the predictive power of the BNICE.ch (*Hatzimanikatis et al., 2005*; *Soh and Hatzimanikatis, 2010*) and BridgIT (*Hadadi et al., 2019*) tools, which identify reactive sites, predict catalysis, and annotate reactions with candidate enzymes. NICEdrug.ch applies these tools in the context of drug metabolism and comprises both phases I and II of such metabolism. NICEdrug.ch extends BridgIT by including reactive site-centric comparisons between molecules, which underlie the principles of competitive inhibition. Additionally, NICEdrug.ch allows to evaluate the toxicity of a drug or its metabolic products by identifying molecular structures linked to toxicity. NICEdrug.ch is available as an open-access resource. The outputs of NICEdrug.ch provide comprehensive visualized reports on the reactivity of any organic molecule (including its metabolic fate or degradation and metabolic precursors or prodrugs), the repurposing of drugs, and the assessment of the druggability of enzymes.

*Supplementary file 2* provides an overview of the principles, scope, availability, application, and annotation capabilities of the tools to predict drug metabolism. This comparative analysis demonstrates the ability of NICErug.ch to address the shortcomings of other tools, for example, by improving performance, expanding scope, and allowing accessibility.

## Quantitative comparison of drug–metabolite prediction

Published frameworks that are comparable to NICEdrug.ch's reactivity report include GLORY (*de Bruyn Kops et al., 2021*; *de Bruyn Kops et al., 2019*), BioTransformer (*Djoumbou-Feunang et al., 2019*), XenoNet (*Flynn et al., 2020*), SyGMa (*Ridder and Wagener, 2008*), and other machine-learning based approaches (*Coley et al., 2017*). All of these methods receive a molecule as an input and predict a set of metabolites as putative products of a metabolic reaction or pathway. Unlike NICEdrug.ch however, they do not describe the reaction mechanisms and metabolic pathways used for each prediction.

Here, we compare the metabolites produced from a given input molecule by the different platforms and compare the output to the molecules produced by the NICEdrug.ch reactivity report for the same compound. We used a curated test set of 29 substrates and their experimentally identified

55 products (labeled in public databases). This set was used in the publication of XenoNet (published in 2020), and we compare it against GLORY, BioTransformer, and SyGMa. NICEdrug.ch produced a reactivity report for each of the 29 parent molecules in the list. We then measured the quality of the predictions using a sensitivity measure, as it has been proposed in the other studies (*Flynn et al., 2020*). Sensitivity is defined as the proportion of true positives among all experimentally observed metabolites. A metabolite is counted to be a true positive when it is both predicted and experimentally observed as a metabolic product of the input molecule. The sensitivity values on the same test set for NICEdrug, XenoNet, GLORY, SyGMa, and BioTransformer are 0.96, 0.89, 0.83, 0.74, and 0.72, respectively. Therefore, NICEdrug outperformed the mentioned tools by predicting a higher number of true metabolites, that is 53 of 55 metabolites (96%). The two remaining metabolites that NICEdrug.ch did not predict are as follows:

1. Venetoclax N-oxide, N-oxidized form of venetoclax (an apoptosis regulator).
2. SCHEMBL18637099, reported as O-dealkylation metabolite of crizotinib (an inhibitor of receptor tyrosine kinase).

The conversion of venetoclax to venetoclax N-oxide requires the biotransformation of an aniline to a nitrobenzene ring. To the best of our knowledge, there is currently no known metabolic reaction in public databases explaining this conversion. In this case, the lack of mechanistic knowledge hinders the formulation of enzymatic reaction rules and consequently the prediction of this specific reaction by NICEdrug.ch. In fact, NICEdrug.ch (based on BNICE.ch) intentionally formulates reaction rules only based on known reactions; thus, our enzymatic reaction rules are 100% biologically interpretable.

SCHEMBL18637099 is the product of the O-dealkylation of crizotinib, which consists of a two-step reaction mechanism. The first step is catalyzed by CYP type of enzymes, catalyzing the hydroxylation on a carbon attached to an oxygen, followed by a second spontaneous step, cleaving a C–O bond and producing the final product (SCHEMBL18637099) (*Silverman, 2004*). It is worth mentioning that using NICEdrug.ch, one can predict the first intermediate of the enzymatic step, but predicting mechanism of non-enzymatic reactions remains out of the scope of NICEdrug.ch.

In addition to the structure of predicted downstream drug metabolites, NICEdrug.ch provides information about the pathways, where each reaction is annotated with a candidate enzyme from phase I or II of drug metabolism. Thirty-nine of the 55 reference metabolites are directly produced from the parent drug in a single reaction step. From the remaining metabolites, (1) eight are predicted in two reaction steps, (2) two in three reaction steps, and (3) the last four metabolites are predicted to be produced using four reaction steps (*Supplementary file 2*).

Among the previously mentioned tools, only Biotransformer postulates hypotheses on reactions from drug metabolism. Here, we discuss in detail the performance of Biotransformer and NICEdrug to predict the metabolism of 5-FU compound. 5-FU is an anticancer drug and application of NICEdrug.ch on predicting its downstream toxic metabolites and ways to alleviate its toxicity is discussed in detail in the paper. According to the KEGG database, 5-FU participates in seven metabolic reactions catalyzed by five enzymes present in human cells. One of the metabolic reactions is catalyzed by CYP type of enzymes (EC 1.14.14.1), five other reactions are oxidoreductases and transferases, and the mechanism of the last reaction (explaining the conversion of Carmofur and 5-FU) is not yet characterized, i.e., this reaction remains orphan and incomplete.

Using the online version of Biotransformer, one can find only a novel biotransformation suggested to be catalyzed by a CYP enzyme. NICEdrug.ch takes advantage of a broad coverage of BNICE.ch reaction rules. With such rules, NICEdrug.ch is able to characterize known metabolic reactions and reconstruct all known and non-orphan metabolic reactions involved in metabolism of 5-FU (1CYP and five oxidoreductases and transferases). Moreover, NICEdrug.ch predicts 92 novel metabolic reactions explaining alternative conversion scenarios of 5-FU in a human cell. Specially, one of the novel predicted reactions involves the conversion between Carmofur and 5-FU in which water and hexylcarbamic acid act as cofactor and co-substrate, respectively. To further explore the metabolic vicinity of 5-FU, we predicted two metabolic conversions (steps) away from 5-FU. Biotransformer suggests there are uniquely two novel metabolic reactions (EC 2.4.1.17 and 3.5.2.2) up to two steps away from 5-FU. While NICEdrug.ch identifies a total of 754 reactions in the two-step away metabolic neighborhood of 5-FU. Of the 754 reactions predicted by NICEdrug.ch, 25 reactions are approved in biochemical databases (*Kanehisa et al., 2017*), which suggests NICEdrug.ch is a better

predictor of drug metabolism than Biotransformer. This is because NICEdrug.ch predicts the metabolism of small molecules using catalysis rules derived from known metabolic reactions. These rules additionally allow to account for enzyme promiscuity.

Finally, we close this section by testing the performance of NICEdrug.ch on the prediction of all experimentally validated drug–metabolite pairs as discussed in a review paper (*Kirchmair et al., 2015*) published in Nature Reviews Drug Discovery as a reference list. We assessed the performance of NICEdrug.ch to predict precursor drug or prodrug metabolism and identify drug–metabolite pairs.

The reference list includes 16 drug–metabolite pairs that are (1) prodrugs, (2) drugs with active or highly active metabolites, and (3) metabolites that have comparable or improved therapeutic properties and are marketed as drugs. Our comparison of predicted and experimentally tested drug–metabolite pairs shows that NICEdrug.ch is remarkably able to correctly find approved metabolites and pathways of metabolism for 15 (of 16) drugs in the reference list (*Supplementary file 2*). The only remaining drug that was not processable with NICEdrug.ch platform is cisplatin, an <u>inorganic</u> platinum-based small molecule and hence out of the scope of NICEdrug.

## Quantitative comparison of recognizing cytotoxicity in bioassay data

Accurate computational prediction of drug toxicity is a major challenge in drug discovery. The cytotoxicity assays are the best descriptors of in vitro and clinical toxicity (*Webel et al., 2020*; *Yin et al., 2019*). Here, we focused on the collection of cytotoxicity bioassay records from PubChem, which have often been used to evaluate accuracy of the computational tools for this means (*Svensson et al., 2017*; *Webel et al., 2020*; *Yin et al., 2019*). This PubChem dataset includes high-throughput screening results for 1777 drug candidates (test dataset), of which 108 are labeled toxic and the rest are non-toxic. To examine the performance of NICEdrug.ch, we predict the toxicity of the 1777 drug candidates. We compute the NICEdrug.ch toxicity score as defined in the manuscript for each molecule in the test dataset. Afterwards, we compared the performance of NICEdrug.ch and other computational tools (*Svensson et al., 2017*; *Yin et al., 2019*). We use the accuracy as a metric, which describes the % of toxic and non-toxic predictions that match the experimental results from the overall set of prediction (*Figure 8A*).

As one can see, with a threshold of 10% NICEdrug toxicity score, we can correctly identify 76 toxic molecules of 108 approved toxic molecules and identify 1126 non-toxic molecules of 1669 approved non-toxic molecules in the test set. At a threshold of 74%, accuracy reaches a maximum value of 0.94, and in higher thresholds in consequence of rejecting some true positives, accuracy slightly decreases to 0.937 (for a threshold value of 1). Other toxicity prediction tools, based on machine learning (*Svensson et al., 2017*; *Yin et al., 2019*), show an accuracy between 0.67 and 0.78 for the same test dataset as reported before (*Svensson et al., 2017*; *Yin et al., 2019*). Compared to these methods, NICEdrug.ch yields a good prediction quality on molecule toxicity identification, as demonstrated by the high accuracy, high precision, and recall for a threshold of 0.74.

From all molecules that are labeled toxic in the test set, there is uniquely one for which NICEdrug.ch did not predict toxicity (the NICEdrug.ch toxicity score is zero). This molecule is aphidicolin, an antibiotic with antiviral and antimitotical properties. Aphidicolin is a tetracyclic diterpenoid that has a tetradecahydro-8,11a-methanocyclohepta[a]naphthalene skeleton and is defined as a reversible inhibitor of EC 2.7.7.7 (DNA-directed DNA polymerase) (*Hastings et al., 2013*). To compute the toxicity score, NICEdrug.ch identifies toxic alerts in the molecules ('Materials and methods', definition of the toxicity score in NICEdrug.ch). This molecule does not include any known toxic alert, and hence NICEdrug.ch does not predict it as toxic. This is consistent with the result of toxicity prediction tools that are only based on toxic alerts (*Sushko et al., 2012*).

In addition, we searched for aphidicolin in other PubChem bioassays (AID 847 and 364), and we found the results of AID 847 and 364 in contrary show aphidicolin is not toxic. This means, in the specific case of aphidicolin, toxicity analysis remains inconclusive and context dependent.

Finally, we investigated the overall predictive performance of NICEdrug.ch with a receiver operating characteristic (ROC) curve (*Figure 2*). The area under the curve (AUC) is 0.75, indicating good generalized performance on the test dataset.

This analysis shows that NICEdrug.ch is able to alert when a new molecule (or the compounds downstream from its metabolism) shares toxic molecular structures with a known toxic compound.

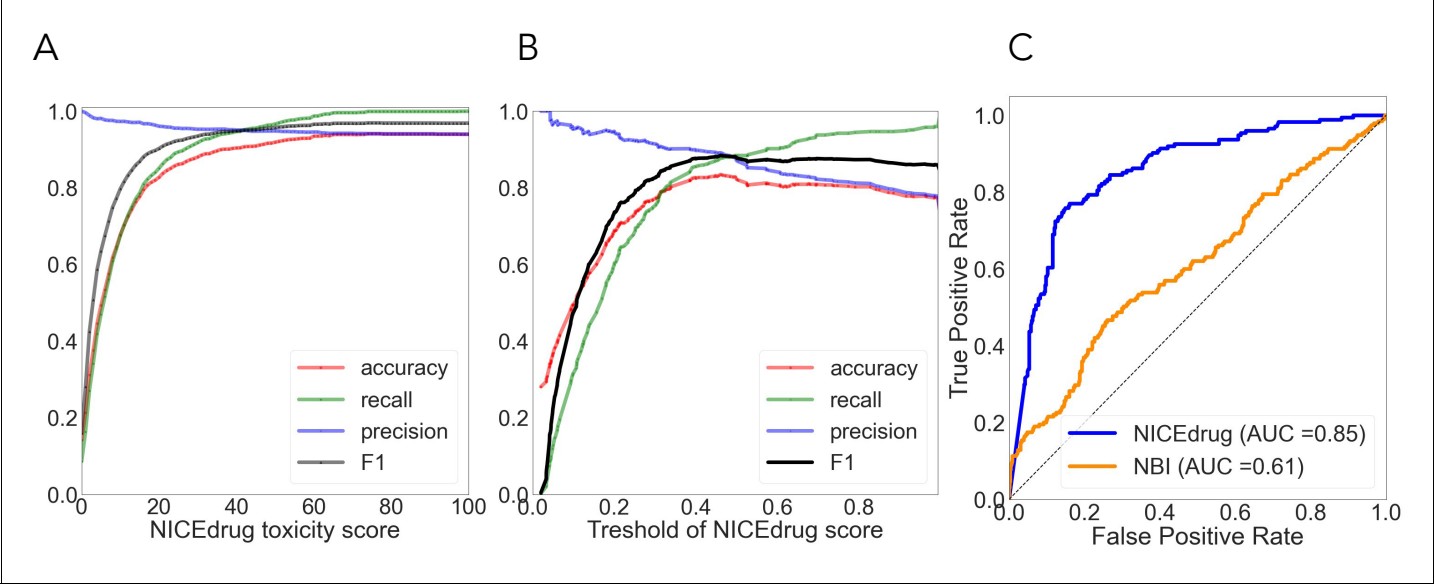

**Figure 8.** Quantitative comparison of drug toxicity (**A**) and drug–enzyme pairs (**B**, **C**). (**A**) Evaluation of NICEdrug.ch toxicity predictions on the test dataset of 1777 drug candidates from PubChem. Y-axis represents the value of a metric, namely accuracy (red), precision (green), recall (blue), and F1 (yellow) based on putative threshold for NICEdrug toxicity score. X-axis represents the predicted NICEdrug.ch toxicity score. (**B**) Evaluation of overall performance of NICEdrug.ch and NBI tool in predicting interaction of enzymes and drug pairs in terms of ROC curves and value of AUC. (**C**) Quality of enzymes and drug pairs predictions using NICEdrug.ch in terms of statistical measures including: precision, recall, F1, and accuracy.

Hence, NICEdrug.ch's annotation of molecules with structural toxic alerts indicates a compound's potential toxic effects.

## Quantitative comparison of identifying drug–enzyme interaction

We compared the predictive performance of NICEdrug.ch with the widely used 'network-based inference (NBI)' tool for DTI prediction (*Cheng et al., 2012b*; *Wu et al., 2020*). NBI predictions are based on drug–target bipartite network topology similarity.

To compare NICEdrug.ch's druggability report with NBI, we gathered a large dataset including the highest quality drug–protein interactions records from PubChem bioassays (*Kim et al., 2021*; *Wang et al., 2012*). We removed from the collected dataset the contrary bioassay records and all bioassays with targets having partial EC annotation, which means it does not contain a four-level EC number-. The final dataset includes 651 bioassay records, reporting inhibition of 78 enzymes by 297 molecules. As a performance assessment criterion, we used the AUC of the ROC curve, which is commonly used as a metric for assessing computational prediction methods (*Mayr et al., 2018*).

We observed that NICEdrug.ch significantly outperforms NBI with an AUC of 0.85, compared to 0.61 for NBI (*Figure 8C*). Compared to the NBI method, NICEdrug.ch only uses the structural information of drug molecules and yields a higher predictive performance, which is remarkable, since other methods that are only based on drug structure have reported lower performance compared to NBI (*Cheng et al., 2012b*; *Wu et al., 2020*). Therefore, this comparison highlights the added value of similarity evaluation centered around reactive sites, which is the backbone of NICEdrug.ch comparisons.

We next analyzed the accuracy, precision, recall, and F1 of NICEdrug.ch predictions as a function of the discrimination threshold (DT), also known as cut off threshold of the NICEdrug score (*Supplementary file 2*). The accuracy ranged from 0.28 (for a DT value of 0.02) to 0.84 (for a DT value of 0.46) (*Figure 8B*). The classification of predictions was overly conservative for DT values greater than 0.46 because it was rejecting true positives. In summary, (1) for DTs between 0 and 0.46, by increasing DT the accuracy increases by reducing the number of false positives, after that (2) for DTs greater than 0.46, the rejection of true positives caused the accuracy to decrease from 0.84 toward 0.72 for a DT value of 1. Based on this analysis, we chose 0.46 as an optimal DT value

for drug–target studies. The values of precision, recall, and F1 corresponding to the optimal DT are 0.88, 0.89, and 0.89 (*Supplementary file 2*).

## Acknowledgements

We would like to thank Dr. Ljubisa Miskovic, Dr. Volker T Heussler, and Dr. Reto Caldelari for critical feedback and discussions. The *Figures 6* and *7*, and their supplements were done with BioRender. Marvin was used for drawing and displaying chemical structures in all figures, Marvin 17.21.0, Chem-Axon (https://www.chemaxon.com). We used morpheus software to generate *Figure 5* (https://software.broadinstitute.org/morpheus). HM was funded by the European Union's Horizon 2020 research and innovation program under the Marie Skłodowska-Curie grant agreement no 72228. HM, KH, JH, and VH were supported by the Ecole Polytechnique Fédérale de Lausanne (EPFL). AC-P and VH were funded by grant 2013/155 (MalarX), and NH and JH were funded by grant 2013/158 (MicroScapesX); both are SystemsX.ch grants, the Swiss Initiative for Systems Biology evaluated by the Swiss National Science Foundation (SNSF). This work has received funding from the Swiss National Science Foundation under grant agreement 200021_188623.

## Additional information

### Funding

| Funder | Grant reference number | Author |
|---|---|---|
| H2020 Marie Skłodowska-Curie Actions | 72228 | Homa MohammadiPeyhani |
| SystemsXch | 2013/155 (MalarX) | Anush Chiappino-Pepe Vassily Hatzimanikatis |
| SystemsX.ch | 2013/158 (MicroScapesX) | Jasmin Hafner Noushin Hadadi |
| École Polytechnique Fédérale de Lausanne | | Homa MohammadiPeyhani Kiandokht Haddadi Jasmin Hafner Vassily Hatzimanikatis |

The funders had no role in study design, data collection and interpretation, or the decision to submit the work for publication.

### Author contributions

Homa MohammadiPeyhani, Conceptualization, Resources, Data curation, Software, Formal analysis, Validation, Investigation, Visualization, Methodology, Writing - original draft, Project administration, Writing - review and editing; Anush Chiappino-Pepe, Conceptualization, Investigation, Visualization, Methodology, Resources, Writing - original draft, Writing - review and editing; Kiandokht Haddadi, Data curation, Software, Formal analysis, Investigation, Methodology, Writing - review and editing; Jasmin Hafner, Software, Writing - review and editing; Noushin Hadadi, Conceptualization, Methodology, Writing - review and editing; Vassily Hatzimanikatis, Conceptualization, Supervision, Funding acquisition, Methodology, Writing - original draft, Project administration, Writing - review and editing

### Author ORCIDs

Homa MohammadiPeyhani https://orcid.org/0000-0002-1308-4659
Anush Chiappino-Pepe https://orcid.org/0000-0002-3993-907X
Kiandokht Haddadi https://orcid.org/0000-0001-5950-9571
Jasmin Hafner https://orcid.org/0000-0002-1474-5869
Noushin Hadadi https://orcid.org/0000-0001-6614-4910
Vassily Hatzimanikatis https://orcid.org/0000-0001-6432-4694

Decision letter and Author response
Decision letter https://doi.org/10.7554/eLife.65543.sa1
Author response https://doi.org/10.7554/eLife.65543.sa2

# Additional files

## Supplementary files

• Supplementary file 1. Information of human metabolism considered in this study, related to *Figures 2–6*. (A) List of cofactors, (B) list of metabolites, and (C) list of EC numbers considered in BNICE.ch for the generation of reactions in the analysis of drug metabolism in a human cell ('Materials and methods').

• Supplementary file 2. Comparing NICEdrug.ch against other prediction tools for drug metabolism and available biochemical assays, related to *Figure 8*. (A) Comparison of available computational tools to predict drug metabolism. (B) Evaluation of NICEdrug.ch potential to predict the druggability (through competitive inhibition) of an enzyme by a small molecule based on available biochemical assays and high-throughput compound screenings results. (C) Detailed comparison of NICEdrug.ch druggability results for tolcapone (a neuropsychiatric agent), pravastatin (a cardiovascular agent), oxfenicine (a vasodilator), dopamine (a cardiovascular agent) and acarbose (an antidiabetic agent) against available biochemical assays. (D) Result of large-scale quantitative comparison of drug-metabolite, toxicity and drug-enzyme predictions using NICEdrug and other drug discovery tools. (E) Literature derived examples demonstrating NICEdrug performance in predicting pathways of drug metabolism.

• Supplementary file 3. Metabolic neighborhood of 5-FU, related to *Figures 2–4*. (1) List of compounds in the 5-FU metabolic neighborhood including up to four reactions or steps away. (2) Description of reactions in the 5-FU metabolic neighborhood including up to four reactions or steps away.

• Supplementary file 4. NICEdrug analysis for all molecules with reactive site of statins in NICEdrug.ch, related to *Figure 5*. (A) Matrix of NICEdrug score between each pair of the whole set of 254 molecules in NICEdrug.ch with reactive site of statins. (B) Description of nine drugs candidates for repurposing to replace statins based on NICEdrug.ch. These drugs can act as competitive inhibitors of HMG-CoA reductase like statins.

• Supplementary file 5. Essential genes or enzymes and linked metabolites in liver-stage *Plasmodium* and a human cell, related to *Figure 6*. (A) List of essential genes and associated reactions in liver-stage *Plasmodium*, as obtained from the study (*Stanway et al., 2019*). (B) List of essential genes and associated reactions in a human cell, as obtained from the study (*Wang et al., 2015*). (C) List of metabolites linked to essential genes in liver-stage *Plasmodium*. (D) List of metabolites linked to essential genes in a human cell.

• Supplementary file 6. Description of drugs, prodrugs, metabolites and enzymes analyzed in the study of malaria, related to *Figure 6*. (A) NICEdrug druggability analysis of essential genes or enzymes in liver-stage *Plasmodium*: all drugs sharing reactive-site centric similarity with the *Plasmodium* metabolites and comparison with human metabolites. (B) NICEdrug druggability analysis of essential genes or enzymes in liver-stage *Plasmodium*: all prodrugs (up to three steps away of 346 drugs) sharing reactive-site centric similarity with the *Plasmodium* metabolites and comparison with human metabolites. (C) Description of drugs and prodrugs identified in the malaria analysis with NICEdrug.ch and validated in the study by *Antonova-Koch et al., 2018* along with their similar *Plasmodium* metabolite and human metabolite.

• Supplementary file 7. Hijacked human enzymes by SARS-CoV-2, and drugs and food-based compounds that can inhibit them based on the NICEdrug score, related to *Figure 7*. (A) Hijacked human proteins by SARS-CoV-2 as identified by *Gordon et al., 2020* with an annotated enzymatic function (EC number), also called here 'SARS-CoV-2 hijacked enzymes'. (B) NICEdrug druggability report for SARS-CoV-2 hijacked enzymes including all NICEdrug small molecules. (C) Best candidate drugs against COVID-19: NICEdrug druggability report for SARS-CoV-2 hijacked enzymes including drugs with NICEdrug score above 0.5 compared to the native human substrate. (D) Summary of NICEdrug

best candidate drugs against COVID-19 and their classification according to the drug category in the KEGG database. (E) NICEdrug druggability report of SARS-CoV-2 hijacked enzymes including prodrugs (up to three steps away of any NICEdrug small molecule) with NICEdrug score above 0.5 compared to the native human substrate. (F) Best candidate food-based molecules against COVID-19: NICEdrug druggability report of SARS-CoV-2 hijacked enzymes including food-based molecules with NICEdrug score above 0.5 compared to the native human substrate. (G) Summary of the NICE-drug best candidate food-based molecules against COVID-19 and their classification according to the fooDB source.

- Supplementary file 8. NICEdrug analysis of inhibitory mechanisms of currently used anti SARS-CoV-2 drugs, related to *Figure 7*. (A) All drug molecules and (B) prodrugs in NICEdrug.ch sharing reactive site with the native substrates of the human enzyme HDAC2 and their NICEdrug score with this substrate. (C) All molecules cataloged in fooDB sharing reactive site with the native substrates of the human enzyme HDAC2 and their NICEdrug score with this substrate. (D) All drug molecules and (E) prodrug molecules in NICEdrug.ch sharing reactive site with the native substrates of the human enzyme ACE2 and their NICEdrug score with this substrate. (F) All molecules cataloged in fooDB sharing reactive site with the native substrates of the human enzyme ACE2 and their NICE-drug score with this substrate. (G) All molecules in NICEdrug.ch or cataloged in fooDB sharing reactive site with the native substrates of the human enzyme DNA-directed RNA polymerase and their NICEdrug score with this substrate.

- Transparent reporting form

### Data availability

All data generated or analysed during this study are included in the manuscript and supporting files. Source data files have been provided for all figures (1 to 8). Furthermore, the NICEdrug.ch workflow is implemented in an open-access platform (http://nicedrug.ch/), where users can explore metabolic fate, toxicity, drug repurposing, and enzyme druggability for more than 250k small molecules.

The following previously published datasets were used:

| Author(s) | Year | Dataset title | Dataset URL | Database and Identifier |
|---|---|---|---|---|
| Kim S, Chen J, Cheng T, Gindulyte A, He J, He S, Li Q, Shoemaker BA, Thiessen PA, Yu B, Zaslavsky L, Zhang J, Bolton EE | 2021 | PubChem bioassay data | http://pubchem.ncbi.nlm.nih.gov | PubChem bioassay data, pubchem.ncbi.nlm.nih.gov |
| Wishart D | 2020 | FooDB database | https://foodb.ca/ | FooDB database, foodb.ca/ |
| Kanehisa M, Goto S | 2018 | KEGG database | https://www.genome.jp/kegg/ | KEGG database, www.genome.jp/kegg/ |
| Wishart DS, Knox C, Guo AC, Shrivastava S, Hassanali M, Stothard P, Chang Z, Woolsey J | 2018 | Drugbank database | https://www.drugbank.ca/ | Drugbank, version5.1.6 |
| Gaulton A, Hersey A, Nowotka M, Bento AP, Chambers J, Mendez D, Mutowo P, Atkinson F, Bellis LJ, Cibrián-Uhalte E, Davies M, Dedman N, Karlsson A, Magariños MP, Overington JP, | 2017 | ChEMBL database | https://www.ebi.ac.uk/chembl/ | ChEMBL, CHEMBL24 |

Papadatos G,  Smit
I,  Leach AR

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
