## [Decision Letter]

**Acceptance summary:**

The authors have developed and validated an important tool for the scientific community specifically those interested in drug development and experimental therapeutics. Additional analysis using several comparative analyses on the golden standard sets of drug metabolism, drug toxicities, and drug targets, that further demonstrate the advantage of NICEdrug.ch over similar methods in the field have now been provided. Furthermore, they have now prepared a detailed user-guide that walks through NICEdrug.ch interface step by step using screenshots that visually explain to users how they can navigate the website. In addition, they have addressed the issue of open access to their platform.

**Decision letter after peer review:**

Thank you for submitting your article "NICEdrug.ch: a workflow for rational drug design and systems-level analysis of drug metabolism" for consideration by *eLife*. Your article has been reviewed by 2 peer reviewers, and the evaluation has been overseen by a Reviewing Editor and Mone Zaidi as the Senior Editor. The following individual involved in review of your submission has agreed to reveal their identity: Adil Mardinoglou (Reviewer #1).

As the authors can see there was significant enthusiasm for the work but several of the concerns raised by the reviewers need to be addressed.

Essential Revisions (for the authors):

1) Address the issue raised about the lack of quantitative assessment of the pipeline.

2) Address the issue of usability.

3) Address the open access issue raised by the reviewer.

*Reviewer #1 (Recommendations for the authors):*

Some concerns are listed below:

1. Validation is performed by using PubChem Bioassays database. The authors may include additional literature validation to support their findings.

2. How the authors handle the predictions of enzymes with more than one EC number? Please provide additional information.

3. How the authors handle the discrepancies while collecting information from different databases?

4. "We assumed an enzyme is druggable when the NICEdrug.ch score between the drug and the enzyme's native substrate is above 0.3." Similarly, throughout the study, NICEdrug.ch threshold score was 0.5. How is this 0.3 and thresholds settled? What is the rationale behind this?

5. It is necessary to have no significance control (p-value or FDR) to remove the false positive when predicting drug-enzyme/target pairs. The authors should provide additional information about it.

6. A benchmarking is performed with a chosen tool named Biotransformer, however an extension of comparison with commonly used methodologies would improve the value of this paper.

*Reviewer #2 (Recommendations for the authors):*

1) Two of the most critical drawbacks are, first, the lack of quantitative assessment of the abilities of the service and its analysis pipeline. Use cases provide valuable information; however, it is not possible to assess the overall value of any computational tool/service without large-scale quantitative analyses. One analysis of this kind has been done and explained under "NICEdrug.ch validation against biochemical assays" and "Comparison of NICEdrug.ch predictions and biochemical assays"; however, this is not sufficient as both the experimental setup and the evaluation of results are quite generic (e.g., how to evaluate an overall accuracy of 0.73 without comparing it to other computational methods that produce such predictions, as there are many of them in the literature). Also, similar quantitative and data-driven evaluations should be made for other sections of the study as well.

2) The second critical issue is that, in the manuscript, the emphasis should be on NICEdrug.ch, since most of the underlying computational methods have already been published. However, the authors did not sufficiently focus on how the service can actually be used to conduct the analysis they mention in the use cases (in terms of usability). Via use cases, authors provide results and its biological discussion (which actually is done very well), but there is no information on how a potential user of NICEdrug.ch (who is not familiar with this system before and hoping to get an idea by reading this paper) can do similar types of analyses. I recommend authors to support the textual expressions with figures in terms of screenshots taken from the interface of NICEdrug.ch at different stages of doing the use case analyses being told in the manuscript. This will provide the reader with the ability to effectively use NICEdrug.ch.

3) "NICEdrug.ch identifies toxic alerts in the anticancer drug 5-FU and its products from metabolic degradation."

This is quite nice and informative as a case study; however, to indicate the effectiveness of NICEdrug.ch in identifying toxic alerts in general (rather than on just drug), I recommend authors to carry out a quantitative analysis using a tox-based benchmark dataset and calculate the performance of their method on this dataset, and discuss the results (this can be read as a part of issue 1).

4) "Furthermore, we filtered molecules based on Lipinski rules (Lipinski et al., 2001)"

This filtering operation probably causes the discarding of many drugs. For example, there are roughly 10000 unique small molecule drug entries in DrugBank (investigational/experimental + approved). Here, only 3716 have been selected. It is also known that some of the approved drugs do not obey Lipinski rules. So, this application probably filtered out those approved drugs. Wouldn't it be better to keep, at least, all approved drugs in the database of NICEdrug.ch and to provide the functionalities of the service for those molecules? I presume many users would be interested to query those drugs.

5) Requesting the "Molecule NICEdrug ID" to start a query is highly impractical. Each time the user should go and check the id, then come back to paste it to the box to start the search. I highly recommend authors to add the functionality of searching with name, well-known database ids, and SMILES/InChIKey in these fields.

6) In today's scientific research, open science is one of the most important aspects, and sharing tools, datasets and scripts/source codes/implementations is an important part of it. I observed that authors share their studies/findings/methods over a nicely prepared website/service, considering both the included functionalities and usability; however, requesting registration for letting people use the resource, and doing this via writing an email to the team is not a good practice in my opinion. Many journals that support the open-science movement do not allow the registration requirements for the tools/databases/services published in their journal.

I believe authors have done this just to yield control over who uses their service (i.e., for allowing academic researchers to use the services/tools/databases freely, but not allowing commercial users). However, I believe this is not a good solution. I recommend authors to let access to NICEdrug.ch without the requirement of registration.

---

## [Author Response]

Essential Revisions (for the authors):1) Address the issue raised about the lack of quantitative assessment of the pipeline.

We thank the editor and reviewers for this suggestion. Below we provide new quantitative assessments related to the three main analyses included in NICEdrug.ch.

In brief, NICEdrug.ch performs three main types of analyses: (1) it evaluates the reactivity of any small molecule, which includes predicting its metabolic fate or degradation and its toxicity; (2) it suggests the repurposing of drugs, and (3) it assesses the druggability of enzymes.

It is important to note that no other platform provides all of these analyses. Hence, below we evaluate each analysis separately, and describe how the NICEdrug.ch predictions compare quantitatively with other tools.

Reactivity Report. Published frameworks that are comparable to NICEdrug.ch’s reactivity report include GLORY (de Bruyn Kops et al., 2021, 2019), BioTransformer (Djoumbou-Feunang et al., 2019), XenoNet (Flynn et al., 2020), SyGMa (Ridder and Wagener, 2008), and other machine-learning based approaches (Coley et al., 2017). All of these methods receive a molecule as an input and predict a set of metabolites as putative products of a metabolic reaction or pathway. Unlike NICEdrug.ch however, they do not describe the reaction mechanisms and metabolic pathways used for each prediction. Here, we compare the metabolites produced from a given input molecule by the different platforms, and compare the output to the molecules produced by the NICEdrug.ch reactivity report for the same compound. We used a curated test set of 29 substrates and their experimentally identified 55 products (labeled in public databases). This set was used in the publication of XenoNet (published in 2020), and we compare it against GLORY, BioTransformer and SyGMa. NICEdrug.ch produced a reactivity report for each of the 29 parent molecules in the list. We then measured the quality of the predictions using a sensitivity measure, as it has been proposed in the other studies (Flynn et al., 2020). Sensitivity is defined as the proportion of true positives among all experimentally observed metabolites. A metabolite is counted to be a true positive when it is both predicted and experimentally observed as a metabolic product of the input molecule. The sensitivity values on the same test set for NICEdrug, XenoNet, GLORY, SyGMa, and BioTransformer are 0.96, 0.89, 0.83, 0.74, and 0.72,

respectively. Therefore, NICEdrug outperformed the mentioned tools by predicting a higher number of true metabolites, that is 53 out of 55 metabolites (96%). The 2 remaining metabolites that NICEdrug.ch did not predict are:

1) Venetoclax N-oxide, N-oxidized form of venetoclax (an apoptosis regulator).

2) SCHEMBL18637099, reported as O-dealkylation metabolite of crizotinib (an inhibitor of receptor tyrosine kinase).

The conversion of venetoclax to venetoclax N-oxide, requires the biotransformation of an aniline to a nitrobenzene ring. To the best of our knowledge, there is currently no known metabolic reaction in public databases explaining this conversion. In this case, the lack of mechanistic knowledge hinders the formulation of enzymatic reaction rules and consequently the prediction of this specific reaction by NICEdrug.ch. In fact, NICEdrug.ch (based on BNICE.ch) intentionally formulates reaction rules only based on known reactions, thus our enzymatic reaction rules are 100% biologically interpretable. SCHEMBL18637099 is the product of the O-dealkylation of crizotinib, which consists of a two-step reaction mechanism. The first step is catalyzed by CYP type of enzymes, catalyzing the hydroxylation on a carbon attached to an oxygen, followed by a second spontaneous step, cleaving a C-O bond and producing the final product (SCHEMBL18637099) (Silverman, 2004). It is worth mentioning that using NICEdrug.ch, one can predict the first intermediate of the enzymatic step, but predicting mechanism of non-enzymatic reactions remains out of the scope of NICEdrug.ch.

In addition to the structure of predicted downstream drug metabolites, NICEdrug.ch provides information about the pathways, where each reaction is annotated with a candidate enzyme from phase I or II of drug metabolism. 39 out of the 55 reference metabolites are directly produced from the parent drug in a single reaction step. From the remaining metabolites (i) 8 are predicted in 2 reaction steps, (ii) 2 in 3 reaction steps, and (iii) the last 4 metabolites are predicted to be produced using 4 reaction steps (Supplementary file 2).

Toxicity report. Accurate computational prediction of drug toxicity is a major challenge in drug discovery. The cytotoxicity assays are the best descriptors of in vitro and clinical toxicity (Webel et al., 2020; Yin et al., 2019). Here, we focused on the collection of cytotoxicity bioassay records from PubChem, which have often been used to evaluate accuracy of the computational tools for this means (Svensson et al., 2017; Webel et al., 2020; Yin et al., 2019). This PubChem dataset includes high-throughput screening results for 1,777 drug candidates (test dataset), out of which 108 are labeled toxic and the rest are non-toxic. To examine the performance of NICEdrug.ch, we predict the toxicity of the 1,777 drug candidates. We compute the NICEdrug.ch toxicity score as defined in the manuscript for each molecule in the test dataset. Afterwards, we compared the performance of NICEdrug.ch and other computational tools (Svensson et al., 2017; Yin et al., 2019). We use the accuracy as a metric, which describes the % of toxic and non-toxic predictions that match the experimental results from the overall set of prediction (Author response image 1).

**Author response image 1. respfig1:** Evaluation of NICEdrug. ch toxicity predictions on the test dataset of 1,777 drug candidates from PubChem. Y-axis represents the value of a metric, namely accuracy (red), precision (green), recall (blue), and F1 (yellow) based on putative threshold for NICEdrug toxicity score. X-axis represents the predicted NICEdrug.ch toxicity score.

As one can see, with a threshold of 10% NICEdrug toxicity score, we can correctly identify 76 toxic molecules out of 108 approved toxic, and identify 1,126 non-toxic molecules out of 1,669 approved non-toxic molecules in the test set. At a threshold of 74%, accuracy reaches a maximum value of 0.94, and in higher thresholds in consequence of rejecting some true positives , accuracy slightly decreases to 0.937 (for a threshold value of 1). Other toxicity prediction tools, based on machine learning (Svensson et al., 2017; Yin et al., 2019), show an accuracy between 0.67 and 0.78 for the same test dataset as reported before (Svensson et al., 2017; Yin et al., 2019). Compared to these methods, NICEdrug.ch yields a good prediction quality on molecule toxicity identification, as demonstrated by the high accuracy, high precision and recall for a threshold of 0.74. From all molecules that are labeled toxic in the test set, there is uniquely one for which NICEdrug.ch did not predict toxicity (the NICEdrug.ch toxicity score is zero). This molecule is aphidicolin, an antibiotic with antiviral and antimitotical properties. Aphidicolin is a tetracyclic diterpenoid that has a tetradecahydro-8,11amethanocyclohepta[a]naphthalene skeleton and is defined as a reversible inhibitor of EC 2.7.7.7 (DNA-directed DNA polymerase) (Hastings et al., 2013). To compute the toxicity score, NICEdrug.ch identifies toxic alerts in the molecules (supplementary material, definition of the toxicity score in NICEdrug.ch). This molecule does not include any known toxic alert, and hence NICEdrug.ch does not predict it as toxic. This is consistent with the result of toxicity prediction tools that are only based on toxic alerts (Sushko et al., 2012).

In addition, we searched for aphidicolin in other PubChem bioassays (AID 847 and 364), and we found the results of AID 847 and 364 in contrary show aphidicolin is not toxic. This means, in the specific case of aphidicolin toxicity analysis would be context dependent and inconclusive. Finally, we investigated the overall predictive performance of NICEdrug.ch with a Receiver Operating Characteristic (ROC) curve (Figure 2). The Area Under the Curve (AUC) is 0.75, indicating good generalized performance on the test dataset.

This analysis shows that NICEdrug.ch is able to alert when a new molecule (or the compounds downstream from its metabolism) shares toxic molecular structures with a known toxic compound. Hence, NICEdrug.ch’s annotation of molecules with structural toxic alerts indicates a compound’s potential toxic effects.

Druggability report. We compared the predictive performance of NICEdrug.ch with the widely used “network-based inference (NBI)” tool for Drug-Target interaction (DTI) prediction (Cheng et al., 2012; Wu et al., 2020). NBI predictions are based on drug-target bipartite network topology similarity.

To compare NICEdrug.ch’s druggability report with NBI, we gathered a large dataset including the highest quality drug-protein interactions records from PubChem bioassays (Kim et al., 2021; Wang et al., 2012). We removed from the collected dataset the contrary bioassay records and all bioassays with targets having partial EC annotation, which means it does not contain a 4-level EC number-. The final dataset includes 651 bioassay records, reporting inhibition of 78 enzymes by 297 molecules. As a performance assessment criterion, we used the AUC of the ROC curve, which is commonly used as a metric for assessing computational prediction methods (Mayr et al., 2018).

We observed that NICEdrug.ch significantly outperforms NBI with an AUC of 0.85, compared to 0.61 for NBI (in the manuscript, and here Figure 2A). Compared to the NBI method, NICEdrug.ch only uses the structural information of drug molecules and yields a higher predictive performance, which is remarkable, since other methods that are only based on drug structure have reported lower performance compared to NBI (Cheng et al., 2012; Wu et al., 2020). Therefore, this comparison highlights the added value of similarity evaluation centered around reactive sites, which is the backbone of NICEdrug.ch comparisons.

We next analyzed the accuracy, precision, recall, and F1 of NICEdrug.ch predictions as a function of the discrimination threshold (DT), also known as cut off threshold of the NICEdrug score (Supplementary file 2). The accuracy ranged from 0.28 (for a DT value of 0.02) to 0.84 (for a DT value of 0. 46). The classification of predictions was overly conservative for DT values greater than 0.46, because it was rejecting true positives. In summary, (i) for DTs between 0 to 0.46, by increasing DT the accuracy increases by reducing the number of false positives, after that (ii) for DTs greater than 0.46, the rejection of true positives caused the accuracy to decrease from 0.84 toward 0.72 for a DT value of 1. Based on this analysis, we chose 0. 46 as an optimal DT value for drug-target studies. The values of precision, recall, and F1 corresponding to the optimal DT are 0.88, 0.89, and 0.89 (Supplementary file 2).

2) Address the issue of usability.

We thank the reviewer for the feedback. We have included a complete user manual in the main page of NICEdrug platform, that explains how to perform different tasks of compound search, generating reactivity, repurposing and druggability reports along with examples discussed in the manuscript. To directly access user manual, please use the link here: https://lcsb-databases.epfl.ch/pathways/downloads/NICE-Drug/user_manual.pdf.

3) Address the open access issue raised by the reviewer.

Regarding the NICEdrug.ch availability, the use of the website requires a simple registration, which is free and without discrimination to any entities or persons for non-commercial uses. It has been introduced to prevent abuses of our computational resources (and also use of web robots) to maintain the integrity and quality of our website, so that it can better serve the research community. This set-up also complies with certain legal restrictions that are imposed on the use of the data.

Reviewer #1 (Recommendations for the authors):Some concerns are listed below:1. Validation is performed by using PubChem Bioassays database. The authors may include additional literature validation to support their findings.

To address the reviewer’s concern, we will focus on drug-metabolite pairs, that involve drugs and metabolites that share reactivity similarity based on the similarity of their reactive site and surrounding atoms (NICEdrug score). We have included in the revised manuscript all experimentally validated drug-metabolite pairs as discussed in a review paper (Kirchmair et al., 2015) published in Nature Reviews Drug Discovery as a reference list. We assessed the performance of NICEdrug.ch to predict precursor drug or prodrug metabolism and identify drug-metabolite pairs.

The reference list includes 16 drug-metabolite pairs that are (i) prodrugs, (ii) drugs with active or highly active metabolites, and (iii) metabolites that have comparable or improved therapeutic properties and are marketed as drugs. Our comparison of predicted and experimentally tested drug-metabolite pairs shows that NICEdrug.ch is remarkably able to correctly find approved metabolites and pathways of metabolism for 15 (out of 16) drugs in the reference list (Supplementary file 2). The only remaining drug which was not processable with NICEdrug.ch platform is cisplatin, an inorganic platinum-based small molecule and hence out of the scope of NICEdrug.

2. How the authors handle the predictions of enzymes with more than one EC number? Please provide additional information.

Empowered by the knowledge of the reactive site in the fingerprints of NICEdrug.ch (materials and method: identification of reactive sites in drugs and drug candidates, definition of the NICEdrug score), NICEdrug.ch is the only drug discovery method that can distinguish different EC numbers corresponding to different reaction mechanisms and generate different fingerprints for each EC number. As a consequence, NICEdrug.ch can propose distinct sets of inhibitors corresponding to each distinct mechanism of an enzyme and rank them according to the NICEdrug score. The list of competitive inhibitors can then be prioritized based on each specific EC, NICEdrug ranking, and the host organism.

3. How the authors handle the discrepancies while collecting information from different databases?

Indeed, different applications require different levels, scopes and curation standards, resulting in heterogeneous data resources. In order to obtain consistent and duplicate-free data, we developed a unification algorithm based on canonical SMILES. This procedure allows us to map all properties of molecules across different databases to a unique structure. Basically, compounds were merged using their canonical SMILES as a unique identifier (calculated using openbabel version 2.4.0). To efficiently store, retrieve, and analyze the huge amount of data collected from KEGG, DrugBank and ChEMBL, we created an SQL-based database where we imported the collected data (NICEdrug database). As a result, a unique compound entry in the NICEdrug.ch database can contain different resonance forms, stereoisomers, as well as dissociated and charged states of a same compound. Furthermore, consistent structural descriptors were generated based on the canonical SMILES of each molecule, including the chemical formula, molecular weight, InChI, InChIKey (using openbabel version 2.4.0).

4. "We assumed an enzyme is druggable when the NICEdrug.ch score between the drug and the enzyme's native substrate is above 0.3." Similarly, throughout the study, NICEdrug.ch threshold score was 0.5. How is this 0.3 and thresholds settled? What is the rationale behind this?

The NICEdrug score defines the similarity of two reactive site-centric fingerprints. If two molecules show a high NICEdrug score, there is a high probability that they will both fit inside the enzymatic pocket of the same protein. In a previous study, we introduced the reaction reactive site-centric fingerprint to compare all ligands of a reaction simultaneously, also called BridgIT fingerprint (Hadadi et al., 2019). We compared reactions based on their BridgIT fingerprint and their catalyzing enzymes based the homology of the protein sequence. We then compared the similarity between catalyzing enzymes with the similarity of BridgIT fingerprints for their corresponding reactions. This comparison served to assess how accurately the BridgIT fingerprint predicts protein sequence similarity and putative catalyzing proteins for a reaction. A high similarity between protein sequences and BridgIT fringerprints represented a true positive. We found that with a threshold of 0.3 and 0.5 for the BridgIT fingerprint we reached an accuracy of 0.8 and 0.85, respectively, being the E-value from the BLAST of the protein sequences 10-10 and 10-50 (Hadadi et al., 2019).

This analysis guided the definition of the NICEdrug score based on the reactive site-centric fingerprint. In this manuscript, we defined the threshold of 0.3 for the NICEdrug score to assign similarity in reactivity between two molecules, and 0.5 to identify enzyme targets for a small molecule.

Moreover, the threshold on the NICEdrug score is consistent with the results of large-scale druggability analysis (Supplementary Material, evaluation of the NICEdrug.ch druggability report), where we identified an optimal similarity threshold of 0.46 with values of accuracy, precision, recall and F1 of 0.84, 0.88, 0.89 and 0.89, respectively.

We thank the reviewer for this very important question. To clarify these points, we have added this explanation to the corresponding part of the manuscript (materials and method, How to interpret NICEdrug score).

5. It is necessary to have no significance control (p-value or FDR) to remove the false positive when predicting drug-enzyme/target pairs. The authors should provide additional information about it.

It is worth mentioning that generally, in predictive biochemistry, False Positives are predictions that have not been observed. This means that they might be True Positives, but they have not been reported yet. So, the false discovery rate (FDR) can serve as hypotheses for discovery.

To address this comment, we performed a large-scale comparative study for the identification of drug-target interactions (please see Essential Revisions” number 1 part 3). In this study, we assessed the quality of NICEdrug.ch predictions in terms of drug-enzyme pairs by analyzing statistical features including AUC, accuracy, F1, precision and recall. Here, specifically, we investigated the reviewer’s suggestion by calculating the FDR of drug-enzyme pairs predictions which is equivalent to the ratio of the number of false positive results to the number of total positive test results. As expected, the FDR ranged from 1 (for a threshold value of 0) to 0 (for a threshold value of 1). Based on this analysis, one can select statistically significant predictions with FDR less than 0.05 using a similarity threshold of 0.73 or higher (Supplementary file 2).

6. A benchmarking is performed with a chosen tool named Biotransformer, however an extension of comparison with commonly used methodologies would improve the value of this paper.

Following the reviewer’s suggestion, we have performed a comparative analysis between available tools in the field and NICEdrug.ch. Please see the detailed description in response to “Essential Revisions” number 1.

Reviewer #2 (Recommendations for the authors):1) Two of the most critical drawbacks are, first, the lack of quantitative assessment of the abilities of the service and its analysis pipeline. Use cases provide valuable information; however, it is not possible to assess the overall value of any computational tool/service without large-scale quantitative analyses. One analysis of this kind has been done and explained under "NICEdrug.ch validation against biochemical assays" and "Comparison of NICEdrug.ch predictions and biochemical assays"; however, this is not sufficient as both the experimental setup and the evaluation of results are quite generic (e.g., how to evaluate an overall accuracy of 0.73 without comparing it to other computational methods that produce such predictions, as there are many of them in the literature). Also, similar quantitative and data-driven evaluations should be made for other sections of the study as well.

Following the reviewer’s suggestion, we have performed a comparative analysis between available tools in the field and NICEdrug.ch. Please see the detailed description in response to “Essential Revisions” number 1. This is now also included in the manuscript in the section ‘validation of NICEdrug.ch against biochemical assays’.

2) The second critical issue is that, in the manuscript, the emphasis should be on NICEdrug.ch, since most of the underlying computational methods have already been published. However, the authors did not sufficiently focus on how the service can actually be used to conduct the analysis they mention in the use cases (in terms of usability). Via use cases, authors provide results and its biological discussion (which actually is done very well), but there is no information on how a potential user of NICEdrug.ch (who is not familiar with this system before and hoping to get an idea by reading this paper) can do similar types of analyses. I recommend authors to support the textual expressions with figures in terms of screenshots taken from the interface of NICEdrug.ch at different stages of doing the use case analyses being told in the manuscript. This will provide the reader with the ability to effectively use NICEdrug.ch.

We thank reviewer for this feedback. As it has been discussed in the beginning of this document (response to “Essential Revisions” number 2), we have prepared a detailed user guide, that explains step by step how to reproduce the results of case studies in the manuscript along with screen shots taken from the website. The user guide explains how to choose inputs, how to process them using NICEdrug analysis modules, and how to read the final reports. This document is now available in the main page of NICEdrug.ch (http://nicedrug.ch/), and also directly downloadable from the link provided here:

https://lcsbdatabases. epfl.ch/pathways/downloads/NICE-Drug/user_manual.pdf.

3) "NICEdrug.ch identifies toxic alerts in the anticancer drug 5-FU and its products from metabolic degradation."This is quite nice and informative as a case study; however, to indicate the effectiveness of NICEdrug.ch in identifying toxic alerts in general (rather than on just drug), I recommend authors to carry out a quantitative analysis using a tox-based benchmark dataset and calculate the performance of their method on this dataset, and discuss the results (this can be read as a part of issue 1).

Following the reviewer’s suggestion, we have performed a comparative large-scale analysis on toxicity bioassay datasets. Please see a detailed description in response to “Essential Revisions” number 2. This is now included in the manuscript in the section ‘validation of NICEdrug.ch against biochemical assays’.

4) "Furthermore, we filtered molecules based on Lipinski rules (Lipinski et al., 2001)"This filtering operation probably causes the discarding of many drugs. For example, there are roughly 10000 unique small molecule drug entries in DrugBank (investigational/experimental + approved). Here, only 3716 have been selected. It is also known that some of the approved drugs do not obey Lipinski rules. So, this application probably filtered out those approved drugs. Wouldn't it be better to keep, at least, all approved drugs in the database of NICEdrug.ch and to provide the functionalities of the service for those molecules? I presume many users would be interested to query those drugs.

We thank reviewer for bringing this to our attention. In this study, Lipinski rules have been applied to ChEMBL molecules in an attempt to discard chemicals less likely to show drug properties. This process was not applied on DrugBank molecules and hence has not affected approved drugs. We have now further clarified this point in the manuscript with the following statement under the supplementary material section “Curation of input molecules used in the construction of NICEdrug.ch”:

“Furthermore, we filtered all molecules collected from the ChEMBL based on Lipinski rules in an attempt to discard chemicals less likely to show drug properties (Lipinski et al., 2001):.”

We have carefully reviewed the numbers in our Figure 1—figure supplement 1, which we missed to update with our latest database status. We have now updated Figure 1—figure supplement 1 with the current statistics of NICEdrug.ch compounds based on their source databses. The DrugBank database includes 11,835 drug molecules. To develop NICEdrug.ch, we removed molecules without full structure, curated complex molecules (e.g. salts), canonicalized 2D structures by openbabel version 2.4.0, and eliminated duplicate molecules. To this end, NICEdrug integrates 6,458 unique molecule structures from DrugBank.

5) Requesting the "Molecule NICEdrug ID" to start a query is highly impractical. Each time the user should go and check the id, then come back to paste it to the box to start the search. I highly recommend authors to add the functionality of searching with name, well-known database ids, and SMILES/InChIKey in these fields.

We thank reviewer for this feedback. We have initially separated this process in two steps for two main reasons:

1) The search of all alternative IDs including the NICEdrug ID is extremely fast compared to the other possible analyses. By searching the NICEdrug ID, a user can quickly assess whether the molecule is in the database and prevent obtaining an empty result from other NICEdrug.ch reports if the molecule is not present.

2) It is very important that drugs are correctly looked up and used later for different analysis in NICEdrug.ch, otherwise the whole analysis will be irrelevant. Mapping compounds based on their public identifiers, names and structural descriptors is a laborious process since missing, ambiguous, non-canonical or redundant entries are common among different databases. Furthermore, public identifiers are subject to frequent changes and it is therefore clear that cross-referencing must be done very carefully to enable efficient and reproducible research. We ask researchers to look up their compound of interest in the “compound search” tab of NICEdrug. By doing this, users will first approve the result of compound search before performing any further analysis, by checking its structure, name, SMILE, InCHI, InCHIKey, or public identifiers.

6) In today's scientific research, open science is one of the most important aspects, and sharing tools, datasets and scripts/source codes/implementations is an important part of it. I observed that authors share their studies/findings/methods over a nicely prepared website/service, considering both the included functionalities and usability; however, requesting registration for letting people use the resource, and doing this via writing an email to the team is not a good practice in my opinion. Many journals that support the open-science movement do not allow the registration requirements for the tools/databases/services published in their journal.I believe authors have done this just to yield control over who uses their service (i.e., for allowing academic researchers to use the services/tools/databases freely, but not allowing commercial users). However, I believe this is not a good solution. I recommend authors to let access to NICEdrug.ch without the requirement of registration.

We thank the reviewer for this comment. We have provided a detailed description in response to “Essential Revisions” number 3.